# Estimating the contribution of setting-specific contacts to SARS-CoV-2 transmission using digital contact tracing data

Zengmiao Wang[1,6], Peng Yang[2,3,6], Ruixue Wang[1,6], Luca Ferretti [4,5], Lele Zhao [4,5], Shan Pei[1], Xiaoli Wang[2,3], Lei Jia[2,3], Daitao Zhang[2,3], Yonghong Liu[2,3], Ziyan Liu[1], Quanyi Wang [2,3] ✉, Christophe Fraser [4,5] & Huaiyu Tian [1] ✉

While many countries employed digital contact tracing to contain the spread of SARS-CoV-2, the contribution of cospace-time interaction (i.e., individuals who shared the same space and time) to transmission and to super-spreading in the real world has seldom been systematically studied due to the lack of systematic sampling and testing of contacts. To address this issue, we utilized data from 2230 cases and 220,878 contacts with detailed epidemiological information during the Omicron outbreak in Beijing in 2022. We observed that contact number per day of tracing for individuals in dwelling, workplace, cospace-time interactions, and community settings could be described by gamma distribution with distinct parameters. Our findings revealed that 38% of traced transmissions occurred through cospace-time interactions whilst control measures were in place. However, using a mathematical model to incorporate contacts in different locations, we found that without control measures, cospace-time interactions contributed to only 11% (95%CI: 10%–12%) of transmissions and the super-spreading risk for this setting was 4% (95%CI: 3%–5%), both the lowest among all settings studied. These results suggest that public health measures should be optimized to achieve a balance between the benefits of digital contact tracing for cospace-time interactions and the challenges posed by contact tracing within the same setting.

Contact tracing has been empirically validated as an effective intervention for the containment of infectious disease outbreaks[1], including Ebola[2], SARS[3], MERS[4], and SARS-CoV-2[5–10]. The importance and the guideline for contact tracing were also highlighted by World Health Organization[11]. By December 31, 2022, 197 countries had implemented contact tracing to mitigate COVID-19[12]. With advances in communication technologies, dozens of countries, including Germany[13], United Kingdom[14], United States[15], Australia and New Zealand[16], employed digital contact tracing to complement traditional contact tracing[17,18]. In response to the high transmissibility of SARS-CoV-2 and the occurrence of super spreading events (SSEs), China implemented digital contact tracing that leveraged the health quick-response codes embedded in smartphone for public locations[19,20] during the period of Dynamic zero COVID policy. Digital contact tracing is empirically

[1]State Key Laboratory of Remote Sensing Science, Center for Global Change and Public Health, Faculty of Geographical Science, Beijing Normal University, Beijing, China. [2]Beijing Center for Disease Prevention and Control, Beijing, China. [3]Beijing Research Center for Respiratory Infectious Diseases, Beijing, China. [4]Pandemic Sciences Institute, Nuffield Department of Medicine, University of Oxford, Oxford, UK. [5]Big Data Institute, Li Ka Shing Centre for Health Information and Discovery, Nuffield Department of Medicine, University of Oxford, Oxford, UK. [6]These authors contributed equally: Zengmiao Wang, Peng Yang, Ruixue Wang. ✉e-mail: wangqy@bjcdc.org; tianhuaiyu@gmail.com

effective in preventing transmission and super-spreading events. Importantly, it allowed for the identification of not only the close contacts but also individuals who shared the same space and time, referred to as cospace–time interaction (see Table S1). However, the contributions to transmissions and to the risk of the super spreading events in different settings have not been evaluated due to the lack of systematic tracing and testing of contacts, especially for cospace–time interactions. Additionally, tracing a large number of contacts to find new cases create a huge burden[21,22]. For example, during the Omicron outbreak in Shanghai, China, more than 2.2 million close contacts and secondary contacts were identified[23]. Therefore, the transmission contribution and SSE risk of contact tracing in various settings should be thoroughly assessed[24,25], particularly considering the involvement of a substantial number of individuals in cospace–time interaction.

## Results

The Omicron outbreak in Beijing, China from April 17, 2022 to June 29, 2022 provided a unique opportunity to address this question. Each traced individual underwent multiple PCR tests to confirm their infection status. During this period, a total of 2230 cases (defined based on the positive PCR tests) and 220,878 contacts were identified with detailed epidemiological records (see Table S2–S3), with median number of contacts per case of 58 (IQR 17–158). These contacts encompassed various types of locations, including dwelling, work-place, cospace–time interaction, community settings and unknown settings (see Table S1 for definitions). In total, 1495 transmission pairs were identified, involving 451 infectors and 1495 infectees. Based on the transmission pairs with detailed infection date and symptom onset date, the generation time and incubation period were fitted to Weibull distribution (Fig. S1 and Table S4). When stratified by age and types of location for infectees, our analysis revealed that the largest number of transmission events occurred within the context of cospace–time interactions for all age groups (Fig. 1A). However, variations were observed when the transmission events were classified based on the infectors (Fig. 1B). For age group ≥61, the highest number of trans-mission events (52) occurred in dwelling settings. In the age group 0-20 and 21-60, the cospace–time interactions were associated with the highest number of transmission events. Notably, when stratified by types of location alone, it was observed that 38% of infectees were infected through cospace–time interactions, followed by 26% in dwelling, 15% in workplace, 12% in unknown settings, and 9% in com-munity settings (Fig. 1C). The predominant proportion of cases

attributed to cospace–time interaction suggested their significant role in SARS-CoV-2 transmission. However, the effect of cospace–time interaction may have been amplified by the implemented non-pharmaceutical interventions (NPIs)[26] due to potential sample bias collected during the outbreak (in other words, the control measures can alter the weighting of sampled contacts across different settings, resulting in deviations from a fair sampling of contacts that would represent the natural transmission routes for this infectious disease).

The intensive contact tracing of contacts who tested both positive and negative captured the social contact patterns. After the quality control, 166,546 contacts of traced individuals before isolation or quarantine were left (see Methods for details, Figs. S2, S3). Based on these contacts, the distributions of contact numbers per day of tracing under various locations were fitted (Fig. 2A–E). We acknowledged that age was not considered here for the following reasons: (1) we focused on the role of different locations in SARS-CoV-2 transmission in this study; (2) the sample sizes stratified by locations were small for age group 0-20 and age group ≥61 (Fig. S4); (3) the age distribution of cases closely resembled the overall age demographics in Beijing, hence suggesting a generalized epidemic, and thus the cases were treated as representative of a general population (Table S5). Our findings indi-cated that the gamma distribution provided a robust fit for char-acterizing the contact number per day of tracing under these five locations (Fig. S5). Mean and variance were estimated for dwelling (2.95, 6.45), workplace (13.92, 338.39), cospace–time interaction (29.08, 1198.09), community settings (25.92, 1192.98), and unknown settings (3.62, 13.44). We also adjusted the age distribution in align-ment with the age demographics in Beijing based on subsampling and performed the same analysis. The characteristics of contact patterns did not change significantly a lot (Table S6). These distinct contact number distributions reflected diverse social contact patterns and can be used as baselines for epidemiologically modeling studies.

To evaluate the risk of transmissions under different settings, we extended the mathematical model for infectiousness[5] to incorporate the distributions of contact number per day under different locations. We found heterogeneity across the locations and the time since infection. The most contagious time is 1.25 days since infection under all locations (Fig. S6), consistent with the viral load dynamics of Omi-cron variant (characterized by the probability of being infected with effective contact with a case, reflecting the biological nature of this SARS-CoV-2 variant, Fig. S6). However, the probability of getting

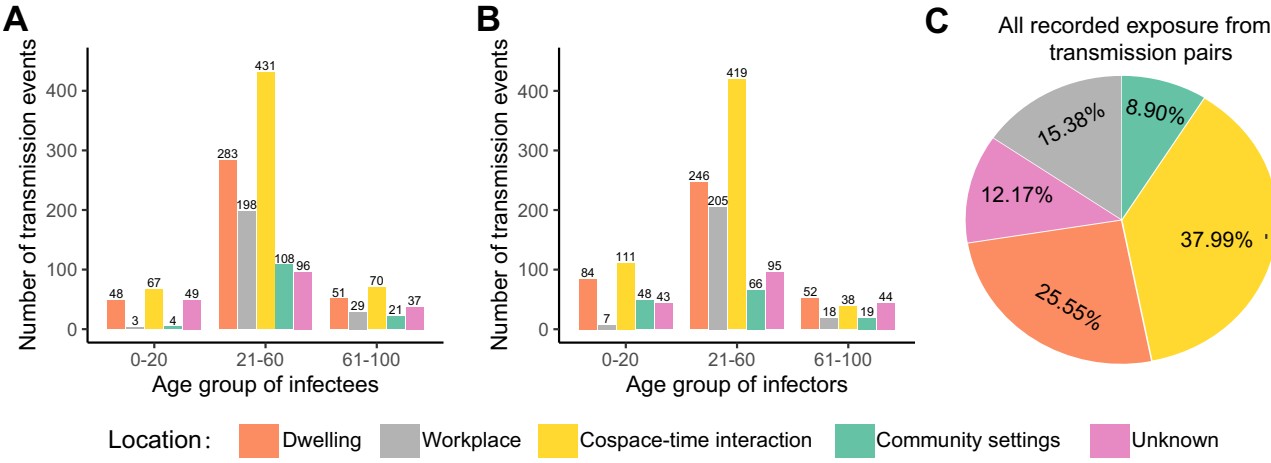

**Fig. 1 | Transmission events stratified by age and locations amidst the Omicron outbreak in the spring of 2022 in Beijing, China. A** Stratification based on age and location of infectees. X-label corresponds to the location and age groups. Y-label corresponds to the counts of transmission event. **B** Stratification based on age and location of infectors. **C** Percentage of transmission events across various locations, derived from systematically, epidemiologically documented 1495 trans-mission pairs. Note that an infector may relate to more than one infectees. There are five different locations: dwelling, workplace, cospace–time interaction, com-munity settings, and unknown settings.

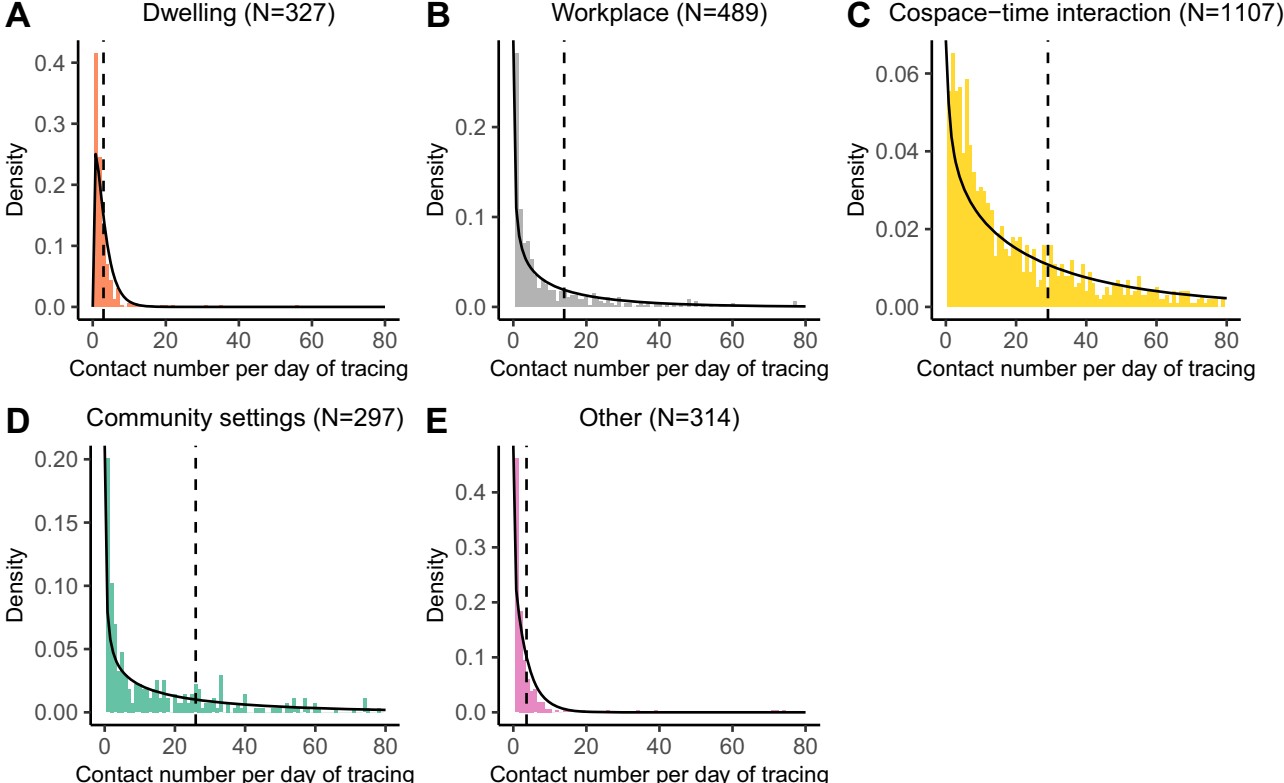

**Fig. 2 | The social contact patterns represented by the contact number per day of tracing across various locations. A–E** The distribution of contact number per day of tracing for an individual under dwelling (**A**), workplace (**B**), cospace–time interaction (**C**), community settings (**D**), and unknown settings (**E**), respectively. Within each subplot, the solid black line represents the gamma distribution curve fitted to contact number per day of tracing, with the vertical dashed line indicating the corresponding mean of the gamma distribution. *N* denotes the number of sample size (i.e., the number of cases) with contact number per day greater than zero under each location. We assume that contacts for these cases in the corresponding location were not influenced by non-pharmaceutical interventions (NPIs) since these contacts were identified before isolation or quarantine. The sensitivity analysis for adjusting the age demographics in Beijing is shown in Table S6.

infected varied for different locations. The largest probability was estimated to be 31% (95%CI: 28–34%) for dwellings, while for cospace–time interactions, it was 0.88% (95%CI: 0.78–0.96%) (Fig. S6). Based on the extended model, the contribution to transmission under dwelling was 40% (95%CI: 35–45%) without NPIs, followed by 17% (95% CI: 14–22%) in workplace, 17% (95%CI: 14–21%) in unknown settings, 14% (95%CI: 11–18%) in community settings and 11% (95%CI: 10–12%) in cospace–time interaction (Fig. 3A). We also evaluated the probability of super spreading event (SSE) under different locations. The results showed that the probability was smallest under cospace–time interaction, which is 4% (95%CI: 3–5%) (Fig. 3B). While cospace–time interactions involved the largest number of contacts (Fig. 2C), they had a relatively low impact on transmission and posed a lower risk of SSE. These findings suggest that cospace–time interactions may not yield significant impact. Considering that the contact patterns might have been influenced by the Dynamic Zero COVID policy during the outbreak, we adjusted the contact patterns based on intra-city mobility data from Beijing (Fig. S7) and conducted the same analysis. While there were some changes, the contribution to transmissions and super-spreading risk of cospace–time interactions did not vary significantly (Fig. S8).

## Discussion
There are several caveats in our study. Generally, a longer exposure time and closer proximity to SARS-CoV-2 increase the risk of transmission[27,28]. Unfortunately, our study did not record the duration of exposure and the distance between cases and contacts. However, the types of locations defined in our study implicitly conveyed this information, despite potential inaccuracies. For the modeling, the independence of the number of contacts among different locations were assumed. Future studies are warranted to investigate the effects of these assumptions. Our analysis primarily focused on epidemiological features at the population level and on a typical day; thus, the contact pattern between weekdays and weekends was not considered. The repeated exposures were not captured in our study and may underestimate the contact number per day, especially for dwelling. However, this wouldn't alter our conclusions, as repeated exposures in dwelling would increase their transmission contribution. A previous study highlighted the impact of heterogeneity in individual infectiousness on transmission control[29]. While our study modeled time-dependent infectiousness on a population level, it did not fully address individual variation. Investigating the roles of both types of heterogeneity is crucial for comprehensive disease control. A more detailed breakdown of cospace–time interactions into the public and enclosed space interactions in the future study would also offer valuable insights for policymaking. Based on our observed data, the likelihood of being infected in cospace–time interaction was low. The low likelihood of being infected would limit the overall transmission contribution attributed to this category, despite the relatively large number of contacts within cospace–time interactions. However, further validation is needed in future studies. In our study, the risk of superspreading events under cospace–time interaction was assessed without control measures. Nevertheless, superspreading varies across space and time[26,30,31], possibly linked to diverse human behaviors and NPIs[32]. Hence, the importance of digital contact tracing for cospace–time interactions in disease control remains significant. Future studies should focus on achieving a balance between the advantages and disadvantages of such tracing methods.

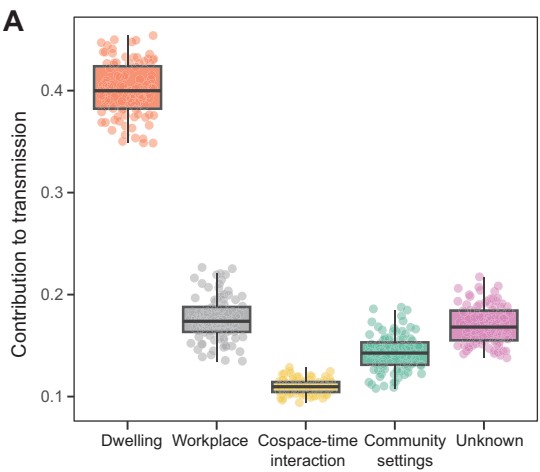
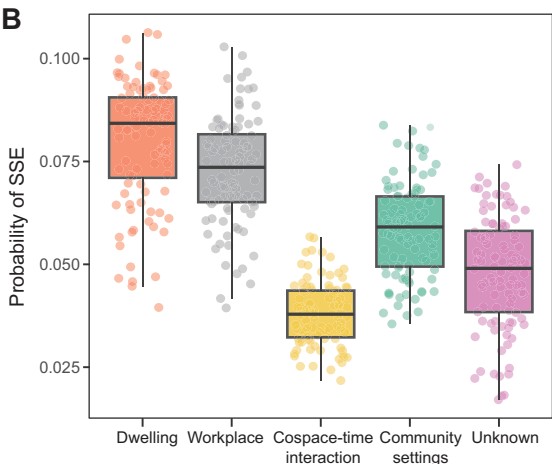

**Fig. 3 | The contributions to transmission and the probability of super spreading event (SSE) under different locations based on the extended mathematical model for infectiousness. A** Contribution to transmission in various locations, presented as an estimated proportion of the overall transmission. In our model, this represents the contribution of transmission in distinct locations to $R_0$. **B** Super spreading event (SSE) risk assessment across diverse locations. Based on the extended a mathematical model for infectiousness, the basic reproduction number ($R_{loc}$) and the distribution of secondary cases were derived for each location. The SSE was defined as the secondary cases of a case exceeding the 99th percentile $Z(R_{loc})$ of a Poisson distribution with $R_{loc}$ as the mean[40]. The probability of SSE was determined as the integral proportion of the secondary cases exceeding $Z(R_{loc})$ based on the distribution of secondary cases, representing the individual's SSE risk under specific location. The box plot displays the median and interquartile range, with whiskers indicating the range within 1.5 times the interquartile range. Points represents the 100 sub-sampling conducted to align the age distribution of cases with the age demographic in Beijing.

In summary, we characterized the contact number per day of tracing under different locations. These social contact patterns can serve as fundamental knowledges, as they were derived based on the intensive contact tracing with confirmed negative contacts. Informed by these social contact patterns, we also evaluated the contribution of different locations to transmission and super-spreading without control measures. These findings suggest that, in comparison to other locations, cospace–time interactions may have a comparatively lower significance if no NPIs were implemented. Our study may provide valuable insights for future pandemic management.

## Methods

### Epidemiological data

The data collection was exempt from Institutional Review Boards, because it was part of a public health investigation for COVID-19, issued by the National Health Commission of the People's Republic of China. This dataset covered 2230 reported SARS-CoV-2 cases (defined based on the positive PCR tests) from April 17, 2022, to June 29, 2022. It included demographic information such as the infectee's list number (assigned due to privacy concerns), gender, age, date of first positive PCR test, infector's list number (assigned due to privacy concerns), infection date for the infectee, reported date for the infectee and isolation date for the infectee. There were missing values in this dataset, especially for the infectors. Ethical approval for this data analysis was provided by the ethical review board of Center for Global Change and Public Health.

### Contact tracing data

Digital contact tracing, facilitated by health quick-response codes (QR codes) embedded in smartphones for public locations, was used to complement traditional contact tracing. The citizens filled their personal information on a sign-up page to obtain a health code[33,34]. They would be assigned a QR code based on the verified information. Each public place was mandated to have a unique QR code, and individuals who entering these places were required to scan the QR codes. The digital contact tracing system recorded the specific time and information of entering citizens. This control measure was employed on December 3, 2021 (source:

https://new.qq.com/rain/a/20211203A03EWM00). To ensure compliance, the corresponding agencies conducted regular inspections of these public places regularly. Additionally, personnel stationed at each entry point checked the health status (green, red or yellow) of individuals scanning the QR code. If the health status were not green, the person was not allowed to enter the place. While these measures cannot guarantee 100% compliance with QR code scanning, we believe the adherence rate to be very high.

The tracing window criteria employed in real world were as follows: (1) Typically, the tracing window spans from 4 days prior to symptom onset or the date of positive sample collection to the day before isolation or quarantine begins; (2) For the primary cases (defined as cases that initiated a contact-tracing process) with an unclear source of exposure, the tracing window may be extended based on the risk assessment, ranging from >4 days (up to 14 days) prior to symptoms onset or the date of positive sample collection to the day before isolation or quarantine begins; (3) For the primary cases with a clearly identified source of exposure and prompt isolation, the tracing window may be shortened to minimize disruption to citizens' daily lives. For each identified SARS-CoV-2 case, the traditional and digital contact tracing was employed[35]. The locations visited by the infected individuals were identified, and others who entered the same locations within the subsequent three hours were identified.

The contact tracing window varied among the cases based on above criteria. In our study, we defined this tracing window as the period from the first to the last day with recorded contacts. For days without recorded contacts, we couldn't differentiate whether tracing was conducted with no contacts found or if the day fell outside the tracing window defined by above criteria. Our defined window implies normal activity during this period. The median of tracing window in our dataset was 3 days with 95% CI: 1–11 (Fig. S2).

The contact tracing dataset comprised 224,323 recorded contacts traced (combing the contacts identified by traditional and digital contact tracing) for 1555 primary cases (defined as cases that initiated a contact-tracing process) identified in the Omicron outbreak. For 675 (675 = 2230−1555) cases, there were no recorded contacts for them. This dataset documented comprehensive information regarding contacts, encompassing the case's list number (assigned due to privacy

concerns), contact mode (including: entertainment, work, living together, caregiving, sharing an elevator, ride-sharing, dining together, same apartment building and others), relationship between the contact and the primary case, contact anonymized ID (assigned by due to privacy concerns), date of last exposure, date of contact's isolation, case's list number if a contact turned positive during the monitoring period. The contacts would undergo 10 days of medical observation followed by 7 days of home observation, during which regular nucleic acid testing and antigen self-testing would be conducted according to the specified guidelines.

### Classification of contact locations
In the context of contact tracing information related to recorded contacts, data cleaning and screening process was carried out by utilizing the attributes of "contact mode" and "relationship between the contact and the primary case". The contact locations were systematically classified into five categories: dwelling, workplace, cospace–time interaction, community settings and unknown settings (Table S1).

### Quality control for the contacts
We conducted data cleaning and filtering for 224,323 contact records traced for 1555 cases (see flowchart in Fig. S3). First, contacts with incorrect birthdates, such as those with missing value or unrealistic ages (e.g., ages less than 0), were removed. This resulted in 223,302 remaining contacts. Second, contacts recorded under school (1706) and long-distance travel (718) locations were excluded due to their small sample size and significant impact from non-pharmaceutical interventions (NPIs) during the Omicron outbreak. After this step, there were 220,878 remaining contacts traced for 1552 primary cases (Table S3). We also excluded cases (44) related to school clusters and their contacts (2284) due to the NPIs. Third, contacts exposed on or after isolation date of the corresponding primary case were excluded, leaving 173,118 recorded contacts traced for 1406 cases. Fourth, considering fast isolating and 26 rounds of population-level PCR testing implemented during the outbreak and ensuring high quality of contact data, we retained the contacts traced within the 14 days leading up to the isolation date of the primary case. And there were 172,703 recorded contacts through tracing for 1404 cases. Furthermore, we excluded contacts (6157) associated with primary cases (283) whose tracking duration covered only one day, as the tracking period may be too short to obtain a reliable value for the average contact number per day at the individual level. Finally, the total contacts used for estimating distributions of contact number per day (as shown in Fig. 2) amounted to 166,546 entries, corresponding to 1121 primary cases (see Table S5).

### Fitting distributions for incubation period and generation interval
In the main text, it was highlighted that the contact database comprised a total of 1495 recorded transmission pairs, each of which included detailed information about the transmission locations. The 1495 transmission pairs involved with 451 infectors and 1495 infectees. Note that one infector may be linked to multiple infectees. Within these 1495 transmission pairs, it is noteworthy that only 226 pairs were recorded with the infectee's date of infection and the onset of symptoms. Furthermore, a subset of 48 transmission pairs exhibited comprehensive documentation encompassing both the infectee and infector's date of infection.

Regarding the dataset of 226 transmission pairs mentioned earlier, we calculated the time duration from the infectee's infection to the onset of symptoms. Additionally, for the dataset of 48 transmission pairs, we also calculated the time duration from the infector's infection to the infection of the corresponding infectee. To model the

distributions for incubation period and generation interval, Bayesian methods[36] were employed, utilizing candidate distributions including gamma, weibull, and lognormal distributions. The results consistently indicate that the weibull distribution outperforms other distributions. The selection of the optimal fitting distribution was based on the lowest leave-one-out information criterion (LOO IC) (Table S4 and Fig, S1).

### Fitting distributions for contact number per day of tracing across various locations
For each primary case, we categorized and counted the traced contacts in different locations before the isolation of the case. We also recorded the tracing time window for each case. This allowed us to compute the average of contact number per day for each case in various locations. To estimate the distribution of contact number per day for each location, we gathered the samples, i.e., the corresponding average of contact number per day for each case. The distribution was fitted against these samples, considering various candidate distributions, including exponential, gamma, weibull, and lognormal distributions. The results suggested that the performance was similar across different distributions (Fig. S5). The gamma distribution was chosen based on its statistical properties, particularly with regard to scaling.

### SARS-CoV-2 transmission model incorporating various contact locations
A previous study developed a mathematical model for infectiousness to evaluate the effectiveness of contact tracing[5]. In this study, we extended this model to incorporate the contacts under different locations. The framework was similar to that in ref. [5]. For readability, the details were described here.

The transmission rate of an individual, denoted as $\beta(\tau)$, is shown as a function of the amount of time since infection, $\tau$. The expression for $\beta(\tau)$ involves the multiplication of the subsequent two factors:

$$\beta(\tau) = R\omega(\tau) \tag{1}$$

where $R$ denotes the reproduction number, and $\omega(\tau)$ serves as the distribution function characterizing the generational interval. And we also have

$$R = \int_0^\infty \beta(\tau) \mathrm{d}\tau. \tag{2}$$

Generally, $\beta(\tau)$ can be decomposed into contributions from distinct infection categories: (i) transmission caused by asymptomatic individuals; (ii) transmission caused by the pre-symptomatic individuals; (iii) transmission caused by symptomatic individuals. Therefore,

$$\beta(\tau) = p_a\beta_a(\tau) + (1-p_a)(1-s(\tau))\beta_p(\tau) + (1-p_a)s(\tau)\beta_s(\tau) \tag{3}$$

where $p_a$ denotes the fraction of asymptomatic cases, $\beta_a(\tau)$ represents the transmission rate of asymptomatic individuals at time $\tau$ post-infection, $s(\tau)$ represents the cumulative distribution function of the incubation period, $\beta_p(\tau)$ represents the transmission rate of pre-symptomatic individuals at time $\tau$ post-infection, and $\beta_s(\tau)$ represents the transmission rate of symptomatic individuals at time $\tau$ post-infection. Our study introduces simplifying assumptions that the infectiousness of asymptomatic individuals and pre-symptomatic individuals is directly proportional to that of symptomatic individuals. Here we have

$$\beta_a(\tau) = x_a\beta_s(\tau), \beta_p(\tau) = x_p\beta_s(\tau). \tag{4}$$

Therefore,

$$\beta(\tau) = p_a x_a \beta_s(\tau) + (1 - p_a)(1 - s(\tau))x_p \beta_s(\tau) + (1 - p_a)s(\tau)\beta_s(\tau). \quad (5)$$

To address the heterogeneity in transmission across diverse contact locations, our study employed a more sophisticated formula:

$$\beta_s(\tau) = \beta_s^{c1}(\tau) + \beta_s^{c2}(\tau) + \beta_s^{c3}(\tau) + \beta_s^{c4}(\tau) + \beta_s^{c5}(\tau) \quad (6)$$

Within this framework, $\beta_s^{c1-c5}(\tau)$ sequentially represents the transmission rate of symptomatic individuals in five locations of contact: dwelling, workplace, cospace–time interaction, community settings and unknown settings. Note that we assumed the independence of transmissions among different locations. Hence,

$$
\begin{aligned}
\beta(\tau) = & \, p_a x_a \beta_s^{c1}(\tau) + (1 - p_a)(1 - s(\tau))x_p \beta_s^{c1}(\tau) + (1 - p_a)s(\tau)\beta_s^{c1}(\tau) \\
& + p_a x_a \beta_s^{c2}(\tau) + (1 - p_a)(1 - s(\tau))x_p \beta_s^{c2}(\tau) + (1 - p_a)s(\tau)\beta_s^{c2}(\tau) \\
& + p_a x_a \beta_s^{c3}(\tau) + (1 - p_a)(1 - s(\tau))x_p \beta_s^{c3}(\tau) + (1 - p_a)s(\tau)\beta_s^{c3}(\tau) \\
& + p_a x_a \beta_s^{c4}(\tau) + (1 - p_a)(1 - s(\tau))x_p \beta_s^{c4}(\tau) + (1 - p_a)s(\tau)\beta_s^{c4}(\tau) \\
& + p_a x_a \beta_s^{c5}(\tau) + (1 - p_a)(1 - s(\tau))x_p \beta_s^{c5}(\tau) + (1 - p_a)s(\tau)\beta_s^{c5}(\tau)
\end{aligned}
\quad (7)
$$

The transmission rate in distinct locations can be decomposed into the product of the following two factors:

$$\beta_j^{ci}(\tau) = c_i p_j^{ci}(\tau), \quad (8)$$

where $i$ represents the category of contact location, $j$ represents transmission category (i.e., asymptomatic, pre-symptomatic or symptomatic), $c_i$ (a value that remains constant over time) denotes the contact number per day for individuals in contact location $i$, and $p_j^{ci}(\tau)$ characterizes the probability of being infected with effective contact under contact location $i$ with a case under $j$ transmission category.

In accordance with formula (2), we can obtain that

$$
\begin{aligned}
R = \int_0^\infty \beta(\tau)d\tau = & \, c_1 \int_0^\infty p_s^{c1}(\tau)\left(p_a x_a + (1 - p_a)(1 - s(\tau))x_p + (1 - p_a)s(\tau)\right)d\tau \\
& + c_2 \int_0^\infty p_s^{c2}(\tau)\left(p_a x_a + (1 - p_a)(1 - s(\tau))x_p + (1 - p_a)s(\tau)\right)d\tau \\
& + c_3 \int_0^\infty p_s^{c3}(\tau)\left(p_a x_a + (1 - p_a)(1 - s(\tau))x_p + (1 - p_a)s(\tau)\right)d\tau \\
& + c_4 \int_0^\infty p_s^{c4}(\tau)\left(p_a x_a + (1 - p_a)(1 - s(\tau))x_p + (1 - p_a)s(\tau)\right)d\tau \\
& + c_5 \int_0^\infty p_s^{c5}(\tau)\left(p_a x_a + (1 - p_a)(1 - s(\tau))x_p + (1 - p_a)s(\tau)\right)d\tau
\end{aligned}
\quad (9)
$$

Integrating each term separately, we will have

$$R = R_{c1} + R_{c2} + R_{c3} + R_{c4} + R_{c5} \quad (10)$$

In this scenario, $R_{c1-c5}$ stands for reproduction numbers in five contact locations: dwelling, workplace, cospace–time interaction, community settings and unknown settings. Assuming that $p_j^{ci}(\tau)$ under different contact locations is directly proportional to the dwelling location with a specific scaling factor, i.e., $y_{ci}$, $R$ can be stated as

$$
\begin{aligned}
R = & \, (c_1 + c_2 y_{c2} + c_3 y_{c3} + c_4 y_{c4} + c_5 y_{c5}) \\
& \int_0^\infty p_s^{c1}(\tau)\left(p_a x_a + (1 - p_a)(1 - s(\tau))x_p + (1 - p_a)s(\tau)\right)d\tau.
\end{aligned}
\quad (11)
$$

For the probability of being infected with effective contact under contact location $i$, $p^{ci}(\tau)$, we have

$$p^{ci}(\tau) = p_s^{ci}(\tau)\left(p_a x_a + (1 - p_a)(1 - s(\tau))x_p + (1 - p_a)s(\tau)\right) \quad (12)$$

## The contributions to transmission under different locations

In this section, we assumed that the contact number in different locations ($c_i$) were random variables to characterize the contact pattern among population and mutually independent. It's important to note that $c_i$ is a value that remains constant over time. By taking the mathematical expectation of both sides of formula (11), we can obtain

$$
\begin{aligned}
E(R) = & \, R_0 = (E(c_1) + E(c_2)y_{c2} + E(c_3)y_{c3} + E(c_4)y_{c4} + E(c_5)y_{c5}) \\
& \int_0^\infty p_s^{c1}(\tau)\left(p_a x_a + (1 - p_a)(1 - s(\tau))x_p + (1 - p_a)s(\tau)\right)d\tau.
\end{aligned}
\quad (13)
$$

The probability of being infected with effective contact under contact location $i$, $p^{ci}(\tau)$, was our focus (Fig. S6). Next, we described how to obtain them without loss of generality. Similar to Ferretti, Luca et al.[5], our analysis began by numerically solving $\beta_s(\tau)$ from Eq. (5) based on the shape of asymptomatic, pre-symptomatic plus symptomatic contributions to the inferred generation interval distribution. Then, we were able to obtain $p_s^{c1}(\tau)$ using formula (6) and (8) with $\beta_s(\tau)$, $y_{ci}$, and $c_i$. With $p_s^{c1}(\tau)$ determined, $p^{ci}(\tau)$ for other locations can be calculated based on formula (12). The contributions to transmission at different contact locations (i.e., $R_{loc}$, where $loc = c_1,...,c_5$) were also computed based on formula (13). The parameters [$R_0$, $s(\tau)$, $\omega(\tau)$, $p_a$, $x_a$, $x_p$, $y_{ci}$ and $c_i$] were known during this analysis.

In our study, $R_0$ was set to $10^{37}$. $s(\tau)$ and $\omega(\tau)$ were set as the estimations in Table S4 (see section Fitting distributions for incubation period and generation interval). The $p_a$ was set to 21.8% based on data for primary cases. $c_i$ was set based on Fig. 2. To obtained $y_{ci}$, we first determined the proportion of infected contacts under each contact location, calculated as the observed infected contacts divided by all the contacts in the corresponding location in Table S3. Specifically, it is 12.7% for dwelling, 1.2% for workplace, 0.4% for cospace–time interaction, 0.5% for community settings and 4.3% for unknown settings. Taking the dwelling as reference, $y_{c1} = 1$, $y_{c2} = 1.2\%/12.7\% = 0.094$, $y_{c3} = 0.4\%/12.7\% = 0.031$, $y_{c4} = 0.5\%/12.7\% = 0.039$, $y_{c5} = 4.3\%/12.7\% = 0.339$. The parameters $x_a$ and $x_p$ were set to 1 based on previous studies[5,38,39]. With $x_a = x_p = 1$, solving $\beta_s(\tau)$ was relatively straightforward (this is a very special scenario).

An intensive contact tracing effort was conducted in Beijing, identifying 2230 cases and 220,878 contacts. Notably, our dataset included contacts who tested negative for SARS-CoV-2, providing a unique aspect. Moreover, the contacts before the quarantine or isolation of cases were analyzed. These allow us to capture the social contact patterns, approximating normal contact patterns (Fig. 2). The dynamic infectiousness under different locations (i.e., Fig. S6) were determined under normal contact pattern, which can be treated as biological nature of SARS-CoV-2. Subsequently, we calculated the transmission contributions, representing scenarios without NPIs. Although the dynamic zero COVID policy may impact the transmission contribution and dynamic infectiousness across different locations, its effects should be limited in our study. In future study, the effects of control measures on the transmission contribution can be evaluated by utilizing updated $p_s^{c1}(\tau)$ and $c_i$, especially focusing on time-dependent control measures (e.g., where $c_i$ can reflect the human behavior in response to the control measures).

**The probability of super spreading event (SSE) under different locations**

The contact number per day $c_i$ was estimated to follow a gamma distribution with shape of $k_i$ and scale of $\theta_i$ (Fig. 2). Based on the solution of $p_s^{c1}(\tau)$ and the distributions of $c_i$, the basic reproduction number ($R_{loc}$) and the distribution of secondary cases were derived for each location. Specifically, secondary cases follows a gamma distribution with shape of $k_i$ and scale of $\Delta_i$, where $\Delta_i = \theta_i y_{ci} \int_0^\infty p_s^{c1}(\tau)\left(p_a x_a + (1-p_a)(1-s(\tau))x_p + (1-p_a)s(\tau)\right)\ d\tau$. The SSE was defined as the secondary cases of a case exceeding the 99th percentile $Z(R_{loc})$ of a Poisson distribution with $R_{loc}$ as the mean[40]. The probability of SSE was determined as the integral proportion of the secondary cases exceeding $Z(R_{loc})$ based on the distribution of secondary cases, representing the individual's SSE risk under specific location (Fig. 3B).

### Reporting summary

Further information on research design is available in the Nature Portfolio Reporting Summary linked to this article.

## Data availability

The data supporting the findings of this study are available within the manuscript. Raw contact data and epidemiological data were not publicly available due to privacy concern. Specific academic requests for access to these data should be directed to the corresponding author (wangqy@bjcdc.org) and Beijing Center for Disease Prevention and Control (https://www.bjcdc.org.cn/). The request will be evaluated within one month.

## Code availability

Codes are available from GitHub (https://github.com/NinaAvailale/contact-tracing-main).

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

## Acknowledgements

Z.W. acknowledges funding from National Science and Technology Major Project (2021ZD0111201) and National Natural Science Foundation of China (82204160). H.T. is supported by Fundamental Research Funds for the Central Universities (2233300001), National Natural Science Foundation of China (82073616), Capital's Funds for Health Improvement and Research (2022-1G-3014), Beijing Science and Technology Planning Project (Z221100007922019), Beijing Advanced Innovation Program for Land Surface Science (110631111), Key research projects of Beijing Natural Science Foundation-Haidian District Joint Fund (L212056), Research on Key Technologies of Plague Prevention and Control in Inner Mongolia Autonomous Region (2021ZD0006), BNU-FGS Global Environmental Change Program (No. 2023-GC-ZYTS-11), and the Beijing Natural Science Foundation (L232014). The funders had no role in study design, data collection and analysis, the decision to publish, or in preparation of the manuscript.

## Author contributions

H.T. and Z.W. conceived of the presented study. P.Y., X.W., L.J., D.Z., and Y.L. collected data for this study. Z.W. and R.W. conducted the analysis. L.F., L.Z., and Z.L. assisted with analysis. Z.W., R.W., and H.T. drafted the manuscript. S.P., P.Y., L.F., L.Z., Q.W., and C.F. provided critical comments on this work. All authors edited and revised the manuscript. H.T. and Q.W. supervised this work.

## Competing interests

The authors declare no competing interests.
