## [Peer Review File · Nature Communications]

Estimating the contribution of setting-specific contacts to SARS-CoV-2 transmission using digital contact tracing dataREVIEWER COMMENTS

Reviewer #1 (Remarks to the Author):

How many transmissions of SARS-CoV-2 through the cospace-time interaction is an important question in designing the policy for epidemic prevention, especially adopting the digital contact tracing to complement traditional contact tracing. Wang et al. analyzed the data from 2,230 infections and 220,878 contacts with detailed epidemiological information collected by the digital contact tracing system during the Omicron outbreak in Beijing in 2022. They presented that 37.99% of infections occurred through cospace-time interactions with control measures. By leveraging the daily contact distribution in different locations, the authors developed a mathematic model and found that the cospace-time interactions contributed to only 11.09% of transmission and 3.85% of super-spreading risk without control measures, ranking the lowest among all locations studied.

These results are of some interests and may provide some insights for the policy makers when implementing the digital contact tracing. I have one major concern. As the main information that this manuscript intends to convey is that the cospace-time interactions contribute the lowest fraction of SARS-CoV-2 transmissions and super-spreading risk without control measures. However, these results are estimated by the developed mathematic model, which is with many assumptions. I am not convinced for results estimated by the current model. Is it possible to develop a more realistic model to estimate the transmission contribution of the cospace-time interaction? Also, given these assumptions in the model, is there a confident interval for the estimate of the transmission contribution and the super-spreading risk of the cospace-time interactions.

Besides, the title of this manuscript is "Revisiting the effectiveness of digital contact tracing for cospace-time interactions". I think this study is more related to the estimate of transmission contribution and the super-spreading risk of cospace-time interactions. There are many factors, such as the fraction of individuals with the smart phones installed with the app, the fraction of individual complying the rules of digital contact tracing, etc., determining the effectiveness of digital contact tracing.

Some other comments are also listed as follows

- 1) The authors adopted the contacts of traced individuals before their isolations or quarantines to estimate the distribution of daily contact number when without the control measures. Firstly, there is the social distance policy in China. Even before the isolation, their normal contact patterns are impacted. Secondly, the contact pattern will be different during the weekday and weekend, which should be counted separately. Thirdly, are all places or locations with the QR code or all individuals will scan the code? These should be included in the analysis or discussed in the end.
- 2) Will the vaccination impact the estimates of the transmission contribution and the super-spreading risk of the cospace-time interactions?
- 3) If the unknown settings are also classified as the cospace-time interactions, what will be the estimates of the transmission contribution and the super-spreading risk of the cospace-time interactions without control measures?
- 4) In lines 87-92, the authors presented three reasons of why that the age was not considered. In specific, "(3) the age distribution of infections closely resembled the overall age demographics in Beijing", however, as presented in Table S5, the fraction of individuals with age larger than 60 in the sample is less than half of the population of Beijing for both the age distributions of the infections with identified contacts and contacts. The sample should be adjusted according to the age demographics in Beijing.

Reviewer #2 (Remarks to the Author):

Very useful work documenting the breakdown of infection based on location and number of contacts. Several comments:

1. "Cospace time interaction" isn't a typical phrase. The authors have provided a description of the

differences between cospace time interaction and community interactions. Furthermore cospace time interaction seems to be a mix of contacts in public settings and enclosed settings. Given that it forms more than 35% of the transmission pairs, perhaps the authors could provide a more granular breakdown of the public Vs enclosed space interactions.

This is important for policy making as the type of settings needs to be articulated to determine the type of public health control measures (e.g. physical distancing if in public Vs improve ventilation if in enclosed space)

2. For the probability of infection of 0.87% in cospace time interaction (line 107) is this before or after control measures. Would be nice to have list of these probabilities before and after control measures.

3. line 65, how was contact tracing done? It's hard to understand 58 contacts were made over how many days.

4. Line 80 do the authors mean "effect" rather than "impact"? How does sample bias result in more cospace time interactions, and hence transmission pairs being detected? More elaboration might be needed and to put the outbreak control policy into context

5 line103, most contagious 1.25 days since infection. How does this translate to days pre symptoms onset? For outbreak control, time of infection is seldom observed hence easier to conduct ops based on time of symptoms

6 line 74, authors excluded contacts associated with infection whose tracking duration covered only one day. Why was this exclusion criteria applied and how many contacts were removed in the process.

7. Code could be made available with mockup data to trial run.

Reviewer #3 (Remarks to the Author):

Review of "Revisiting the effectiveness of digital contact tracing for cospace-time interactions" by Wang et al for Nature Communications

Summary

The authors report on the analysis of a digital contact tracing (DCT) dataset from 1121 COVID-19 index cases (with day of symptom onset) and 166,456 contacts, some of which were found to be infected. From these data, they estimate the incubation period, the generation interval, the distributions of the daily contact rates among five settings, and the probabilities of transmission per contact in each setting. They calculate the contributions of each setting to SARS-CoV-2 transmission, and the probabilities of superspreading events occurring in the five settings. These findings are interpreted in the light of the effectiveness of DCT, by (implicitly) assuming that DCT would be especially useful to prevent superspreading events in the setting of cospace-time interactions, where classic contact tracing cannot be carried out. They conclude that a minor part of all transmission occurs in this setting, and that superspreading events are expected to be rare. Yet, they recommend further study to balance between the benefits and costs of DCT.

General comments

The data and analysis are described in the Supplement. The study describes a unique and valuable dataset that can be used to estimate the contact rates and transmission probabilities for SARS-CoV-2 in different settings. Close reading of the Supplement, and looking at data and results left me with many questions, laid out in detail below. Because of all these questions I cannot judge the validity of methods and results at this moment. However, the dataset itself and the general idea about the data analysis is promising.

Apart from not understanding the technical details in the Supplement, I have difficulty in following the argument from results to conclusions about the effectiveness of DCT. The title and introduction suggest that the analysis assesses the effectiveness of DCT, but that is not really the case. In fact,

the statistical results about contact rates and transmission probabilities are used to discuss the consequence for DCT, involving underlying assumptions about conditions under which DCT would be effective. I think it is interesting to discuss the results in this light, but the underlying assumptions should be made clearer. From what I read between the lines, I think that these implicit assumptions are that DCT is mainly effective in transmission in cospace-time interactions and of most use to prevent superspreading events. In addition to explicitly stating the assumptions/conditions, I would like to understand why they are made.

Detailed comments

A first comment is about the result that the distribution of transmission events by location (figure 1) does not match the distribution of the contributions to transmission by location (figure 3). Intuitively I would expect those to be the same: when 38% of transmission events occur in cospace-time-interaction, why is the contribution of cospace-time-interaction to transmission not just 38%. Also, when I read the model equations in the Supplement where I see that you assume that the infectivity profiles ($\omega(\tau)$, $\beta(\tau)$, $p(\tau)$) are identical for all locations apart from scaling constants, so I would expect this even more. The only part where the locations may differ is in the contributions of asymptomatic, presymptomatic and symptomatic transmission, but I cannot follow that part of the analysis. Anyway, I would like to see an intuitive explanation of why these distributions by location in figures 1 and 3 are not identical.

Suppl, line 42: "locations were extracted", but for which time window before detection of an infected individual?

Suppl, line 47, 48, 54, 55: "de-identification", what is that? Anonymized?

Suppl, line 50: "monitoring period", how long was that?

Suppl, lines 45 and 53 mention 1555 infections and 2230 infections, what is the difference between these numbers? Maybe the 1555 are the infections whose contacts were traced, but the description in lines 53-56 explicitly mention the structure of infector-infectee pairs which suggests that it is about traced individuals. I suggest you write one piece of text that links the information in lines 45-56, and maybe also the steps in the quality control so that all numbers are explained in a single line of explanation.

Suppl, lines 45-50: by contacts you mean contacts found by traditional contact tracing only?

Suppl, line 47: "contact mode", does that refer to the five locations in the analysis?

"Entertainment" seems to be something else.

Suppl, lines 73-77: "we excluded ... covered only one day" seems to result in exclusion of 431 infections (from 1552 to 1121 infections), that seems a lot. I don't understand why one-day observations are unreliable, except when these do not cover complete days. In fact, I think that the analyses is better done on daily numbers of contacts anyway (see below my comment on lines 98-106)

Suppl, line 83: "1495 transmission pairs", does the difference between 1552 and 1495 mean that for 57 infections no infector could be found, or that there were multiple candidate infectors from which it is impossible to select the right one? More general: were all the 1495 infectees in the pairs unambiguously linked to one infector? I can hardly imagine that that is the case, especially not with transmissions in dwellings where I would expect clusters of cases.

Suppl, lines 88-95 and Table S4, estimation of incubation period and generation interval distributions: model selection is done and the Weibull models turn out to fit best. However, the estimated means and variances of the three distributions (Weibull, gamma, lognormal) are very different, which suggests that something goes wrong in the estimation procedure. You would expect that the estimated means and variances are similar to the means and variances directly calculated from the data. Can you provide these as well and/or plot the data as histograms in Figure S1?

Suppl, lines 98-106: The mean number of contacts is calculated per case, and then the means are fitted to distributions. Are these means of just a few days of observation per case around the time of possible onward transmission, or are these based on much longer observation periods starting long before infection? That would be relevant, also in relation to the 431 excluded infections with one day of observation.

Suppl, lines 98-106: The mean number of contacts is calculated per case, and then the means are fitted to distributions. These distributions are used to calculate the probability of a superspreading event (SSE), as the probability of exceeding the Poisson-99% percentile. However, SSEs are single events on a single day, so it is more relevant to know the distribution of contact numbers per day, not the distribution of means. And if you would estimate the distribution of contact numbers per day, then why not include the 431 infections with one day of contacts observed?

Suppl, lines 98-106: A suggestion. In the mathematical model, there is a factor for the transmissibility of asymptomatic, pre-symptomatic and symptomatic individuals. Transmissibility may be related to viral load, but also to contact frequencies which may decrease if people get ill. If I understand correctly, you have symptom onsets in your data and you have asymptomatic infections. Is it possible to estimate contact rates for these three categories separately, especially symptomatic vs not-symptomatic?

Suppl, section 3: I can follow the mathematics of the model, but not how the model is used for inference.

1. From what I understand, the generation interval distribution $\omega(\tau)$ is the sum of the three $\beta(\tau)$ functions of the three infection categories, and the equation in line 135 tells me that $\omega(\tau)$ and $\beta(\tau)$ are different only because of x_p and $s(\tau)$ which makes that $\beta(\tau)$ gets a different weight in the time since infection. If you estimate p_a from the data in Table S2 (19.9% asymptomatic) and $\omega(\tau)$ and $s(\tau)$ are as estimated in Table S4 (please make that clear somewhere!), then the equation in line 135 has 3 unknowns: x_a , x_p , and $\beta_S(\tau)$. How are these estimated? Lines 189-191 only tells that you "solved $\beta_S(\tau)$ through fitting...", but not how this was done exactly. Do you assumed x_a and x_p to be 1? And then, is $\beta_S(\tau)$ calculated numerically? Then better say that you solved $\beta_S(\tau)$ numerically from the equation instead of fitting, which suggests a statistical procedure.

2. The scaling factors y_{ci} are estimated from the proportions of infected contacts per scaling location (lines 191-194), from which I understand that you use the fractions at the bottom of Table S3 (please make that clear!).

3. Then I understand that you use the c_i as the means of the estimated distributions as described in lines 98-106. I think that the means of these distributions should be proportional to the denominators of the probabilities at the bottom of Table S3, but maybe this is not exactly so because the distributions are fitted with means per infection whereas the denominators in Table S3 are the totals. If you would fit the distributions to daily observations (as I suggest), I think the means of the distributions should really become proportional to the denominators in Table S3, so that the product of y_{ci} and c_i will simply reflect the proportions of cases observed in the five settings (numerators at the bottom of Table S3, and Figure 1 main text).

4. In lines 194-199, a large number M is introduced but I don't understand why and what the value would be. Then, equations 5 and 8 are used to solve $p_S^{c1}(\tau)$, but from what I understand from lines 174-177 and eq(8), all functions $p_j^{ci}(\tau)$ are only different by the constants y_{ci} and c_i , the product of which would be the relative contribution to transmission (equation 14). Thus, I don't understand what is meant by solving $p_S^{c1}(\tau)$.

Regarding all points 1-4: maybe I misunderstood what was done, but then I would encourage the authors to explain the calculations/estimations in lines 189-199 in more detail.

Manuscript ID: NCOMMS-23-58901-T

Response to Reviewers

We would like to express our gratitude to the reviewers for their careful review and valuable feedback on our manuscript titled "Revisiting the effectiveness of digital contact tracing for cospace-time interactions" submitted to Nature Communications. This paper is now titled "Estimation of transmission contribution and the super-spreading risk for cospace-time interactions based on digital contact tracing". We appreciate the time and effort that the reviewers dedicated to providing feedback on our manuscript and are grateful for the insightful comments and suggestions on our paper. We have incorporated all the suggestions made by the reviewers. Those changes are highlighted in the manuscript. Please see below, in blue, for a point-by-point response to the reviewers' comments and concerns. All page and line numbers refer to the revised manuscript with tracked changes.

Additionally, the author list has been updated and agreed upon by all authors. Two colleagues were added, Dr. Shan Pei for the independent mathematical model validation and Ziyang Liu for the intra-city mobility analysis.

Please see our responses to the reviewers' specific comments below.

REVIEWER COMMENTS

Reviewer #1 (Remarks to the Author):

How many transmissions of SARS-CoV-2 through the cospace-time interaction is an important question in designing the policy for epidemic prevention, especially adopting the digital contact tracing to complement traditional contact tracing. Wang et al. analyzed the data from 2,230 infections and 220,878 contacts with detailed epidemiological information collected by the digital contact tracing system during the Omicron outbreak in Beijing in 2022. They presented that 37.99% of infections occurred through cospace-time interactions with control measures. By leveraging the daily contact distribution in different locations, the authors developed a mathematic model and found that the cospace-time interactions contributed to only 11.09% of transmission and 3.85% of super-spreading risk without control measures, ranking the lowest among all locations studied.

We appreciate your insights and suggestions, which have helped us improve the quality and clarity of our work. Your input has been instrumental in refining our research and addressing important aspects of the study. We are grateful for your time and consideration in reviewing our manuscript. Please see the detailed responses to the comments below.

These results are of some interests and may provide some insights for the policy makers when implementing the digital contact tracing. I have one major concern. As the main information that this manuscript intends to convey is that the cospace-time interactions contribute the lowest fraction of SARS-CoV-2 transmissions and super-spreading risk without control measures. However, these results are estimated by the developed mathematic model, which is with many assumptions. I am not convinced for results estimated by the current model. Is it possible to develop a more realistic model to estimate the transmission contribution of the cospace-time interaction? Also, given these assumptions in the model, is there a confident interval for the estimate of the transmission contribution and the super-spreading risk of the cospace-time interactions.

Response: Thank you for the valuable feedback. In our study, both digital and traditional contact tracing methods were used to identify the contacts and multiple PCR tests were conducted to confirm the infection status of each contact. By considering the probability of being infected as outlined in Table S3 and the mean contact numbers (Line 120-122 of Page 4 in revised main text) for different locations, we calculated the empirical transmission contributions. For dwelling, the transmission contribution approximately 0.397, calculated as $2.95 \times 12.7\% / (2.95 \times 12.7\% + 13.92 \times 1.2\% + 29.08 \times 0.4\% + 25.92 \times 0.5\% + 3.62 \times 4.3\%)$; for cospace-time interaction, the transmission contribution was about 0.123, calculated as $29.08 \times 0.4\% / (2.95 \times 12.7\% + 13.92 \times 1.2\% + 29.08 \times 0.4\% + 25.92 \times 0.5\% + 3.62 \times 4.3\%)$. Similar methods were used to evaluate the transmission contribution for other locations, and these empirical findings were consistent with our model. In addition, based on the digital contact tracing in England and Wales, the transmissions for household and fleeting was estimated to be 41% and 10%, respectively

(Ferretti L *et al*, Nature, 2024), where household contacts were defined as contacts exposed for at least 8 h in one day and fleeting contacts were defined as contacts exposed less than 30 min. Although fleeting contacts were not directly equivalent to co-space-time interactions, both studies indicate a small transmission contribution from exposures under these circumstances. Despite the relatively small contribution estimated for the co-space-time interactions, it remains important for disease control. We added these into the Discussion section (Line 197-205 of Page 5).

We developed a mathematic model to estimate the fraction of SARS-CoV-2 transmissions and super-spreading risk for the cospace-time interactions. The similar model has been applied in smallpox (Kretzschmar M *et al*, Emerg Infect Dis, 2004) and COVID-19 control (Kretzschmar ME *et al*, Lancet Public Health, 2020). Although our model relies on certain assumptions, its realism is also contingent upon the available data. In our study, the data and the parameters were derived from the real world, ensuring that the results closely approximate reality. The fraction of asymptomatic cases (p_a) was set to 21.8% based on data for primary cases, which was close to previous study (Yu W *et al*, J Med Virol, 2022). The cumulative distribution function of the incubation period $s(\tau)$ and the generational interval $\omega(\tau)$ were set as the estimations in Table S4. x_a and x_p was set to 1 based on previous studies (Ferretti L *et al*, Science, 2020; Wei Z *et al*, Influenza Other Respir Viruses, 2023; Qiu X *et al*, Clin Microbiol Infect, 2021). The scaling factor y_{ci} was estimated based on Table S3. The contact number per day of tracing C_i was estimated based on the contacts data collected in this study. We acknowledged that the assumptions related to the model setup were difficult to relax, representing a necessary simplification process in modeling the complexities of the real world. For example, we assumed constant contact rates [i.e., c_i in equation (9)] over time due to a lack of comprehensive observations throughout the entire disease progression. Additionally, the independence of contacts among different locations were assumed. We added these as limitations in the Discussion section (Line 186-192 of Page 5).

Based on the equation (15) in the revised Supplementary Materials, the primary determinant of the main results is the contact rates under different locations. To address the uncertainty stemming from these contact rates, we performed resampling based on the age demographic in Beijing to estimate the contact rates for these locations and re-calculated the fraction of SARS-CoV-2 transmissions and super-spreading risk for 100 times. Based on the results from re-sampling, the confident intervals were generated. Importantly, the transmission contribution of the cospace-time interactions was still the lowest among all locations studied. We updated Figure 3 in the revised manuscript accordingly.

Ref:

1. Ferretti L, Wymant C, Petrie J, Tsallis D, Kendall M, Ledda A, Di Lauro F, Fowler A, Di Francia A, Panovska-Griffiths J, Abeler-Dörner L, Charalambides M, Briers M, Fraser C. Digital measurement of SARS-CoV-2 transmission risk from 7 million contacts. Nature. 2024 Feb;626(7997):145-150.
2. Kretzschmar M, van den Hof S, Wallinga J, van Wijngaarden J. Ring vaccination and smallpox control. Emerg Infect Dis. 2004 May;10(5):832-41.

3. Kretzschmar ME, Rozhnova G, Bootsma MCJ, van Boven M, van de Wijgert JHHM, Bonten MJM. Impact of delays on effectiveness of contact tracing strategies for COVID-19: a modelling study. *Lancet Public Health*. 2020 Aug;5(8):e452-e459.
4. Yu W, Guo Y, Zhang S, Kong Y, Shen Z, Zhang J. Proportion of asymptomatic infection and nonsevere disease caused by SARS-CoV-2 Omicron variant: A systematic review and analysis. *J Med Virol*. 2022 Dec;94(12):5790-5801.
5. Ferretti L, Wymant C, Kendall M, Zhao L, Nurtay A, Abeler-Dörner L, Parker M, Bonsall D, Fraser C. Quantifying SARS-CoV-2 transmission suggests epidemic control with digital contact tracing. *Science*. 2020 May 8;368(6491):eabb6936.
6. Wei Z, Ma W, Wang Z, Li J, Fu X, Chang H, Qiu Y, Tian H, Zhu Y, Xia A, Wu Q, Liu G, Zhai X, Zhang X, Wang Y, Zeng M. Household transmission of SARS-CoV-2 during the Omicron wave in Shanghai, China: A case-ascertained study. *Influenza Other Respir Viruses*. 2023 Feb;17(2):e13097.
7. Qiu X, Nergiz AI, Maraolo AE, Bogoch II, Low N, Cevik M. The role of asymptomatic and pre-symptomatic infection in SARS-CoV-2 transmission-a living systematic review. *Clin Microbiol Infect*. 2021 Apr;27(4):511-519.

Besides, the title of this manuscript is “Revisiting the effectiveness of digital contact tracing for cospace-time interactions”. I think this study is more related to the estimate of transmission contribution and the super-spreading risk of cospace-time interactions. There are many factors, such as the fraction of individuals with the smart phones installed with the app, the fraction of individual complying the rules of digital contact tracing, etc., determining the effectiveness of digital contact tracing.

Response: We appreciate the constructive suggestion on the title. In this revised manuscript, we have changed it and it is "Estimation of transmission contribution and the super-spreading risk for cospace-time interactions based on digital contact tracing" now. Additionally, we also updated the introduction section correspondingly, as suggested by Reviewer #3. Please check it in Line 60-72 of Page 3.

Some other comments are also listed as follows

1) The authors adopted the contacts of traced individuals before their isolations or quarantines to estimate the distribution of daily contact number when without the control measures. Firstly, there is the social distance policy in China. Even before the isolation, their normal contact patterns are impacted. Secondly, the contact pattern will be different during the weekday and weekend, which should be counted separately. Thirdly, are all places or locations with the QR code or all individuals will scan the code? These should be included in the analysis or discussed in the end.

Response: Thanks for these comments. Firstly, we agree with the reviewer that the contact patterns were still impacted by the Dynamic zero COVID policy implemented at that time. To address this concern, we adjusted the contact pattern based on the intra-city mobility in Beijing. Specifically, the intra-city mobility data suggested that the mobility index in 2023 (after China’s reopening) was,

on average, 1.34 times higher than that in 2022 (Figure S7). We assumed that the impact of social distance policy on contact patterns was proportional to intra-city mobility. Consequently, we multiplied the mean contact patterns for workplaces, cospace-time interactions, community settings, and unknown settings by 1.34. We chose not to adjust the contact pattern for dwellings since these contacts are essential and less likely to be affected by control measures. Based on these adjusted contact patterns, we performed the same analysis. The transmission contribution of cospace-time interactions remained the lowest. However, the super-spreading risk for cospace-time interaction ranked second to lowest among the five locations (Figure S8). Please check them in Line 145-150 of Page 4 in the revised main text.

Secondly, to evaluate the difference of contact patterns between the weekday and weekend, we divided the data accordingly and performed the similar analysis to characterize the contact patterns. The mean and variance of contact number per day of tracing during weekday were estimated for dwelling (3.17, 9.23), workplace (14.44, 379.80), cospace-time interaction (30.10, 1303.15), community settings (25.81, 1182.70), and unknown settings (4.19, 23.62). Similarly, for the weekend, the mean and variance of contact number per day were estimated for dwelling (2.46, 4.08), workplace (10.76, 314.08), cospace-time interaction (26.74, 1513.37), community settings (21.03, 1087.01), and unknown settings (4.06, 30.06). Although there was a trend indicating that contacts on weekends were smaller than those on weekdays, there was no significant difference in contacts between weekdays and weekends (Table R1). Based on this, for the transmission contribution and the super-spreading risk under different locations in the main text, we still used the contacts patterns without differentiation between weekday and weekend. Considering the small difference in contacts between weekdays and weekends and the proportion of weekends in a week, the effects of weekends on our results should be minimal. In addition, our goal was to evaluate these epidemiological features on the population level and on a typical day. The estimated contact patterns without this differentiation can be considered representative of contacts on a typical day. We added these considerations as limitations in the Discussion section (Line 187-189 of Page 5).

Table R1. The contact number per day comparison between before symptom onset and after symptom onset.

Type of locations	Types of contacts	Sample size	Mean	P -value of the t -test
Dwelling	Weekday	n = 250	3.17	0.1193
	Weekend	n = 96	2.46	
Workplace	Weekday	n = 434	14.44	0.1686
	Weekend	n = 173	10.76	
Cospace-time interactions	Weekday	n = 1075	30.10	0.1773
	Weekend	n = 655	26.74	
	Weekday	n = 234	25.81	

Community settings	Weekend	n = 99	21.03	
Unknown settings	Weekday	n = 263	4.19	0.9093
	Weekend	n = 119	4.06	

Thirdly, each public place was mandated to have a unique QR code, and individuals who entering these places were required to scan the QR codes. This control measure was announced by Beijing CDC on December 3, 2021 (source: <https://new.qq.com/rain/a/20211203A03EWM00>). To ensure compliance, the corresponding agencies conducted regular inspections of these public places regularly. Additionally, personnel stationed at each entry point checked the health status (green, red or yellow) of individuals scanning the QR code. If the health status were not green, the person was not allowed to enter the place. While these measures cannot guarantee 100% compliance with QR code scanning, we believe the adherence rate to be very high. We added these in the Supplementary Materials (Line 50-59 of Page 2).

2) Will the vaccination impact the estimates of the transmission contribution and the super-spreading risk of the cospace-time interactions?

Response: Thanks for this constructive comment. We didn't consider the impact of vaccination because Omicron is a SARS-CoV-2 variant with immunity escape ability. About 90% of the cases were breakthrough infection in our dataset (Table S2 in Supplementary Materials). Based on the equation (15) in the revised Supplementary Materials, the relative proportions of transmission contributions among different locations depends on the contact rates since the integral term would be a constant. Therefore, the vaccination would not have an impact on the transmission contributions for different locations. However, the super-spreading risk for different locations may change. Next, we evaluated the impact of vaccination on the super-spreading risk. The vaccination may have impact on the incubation period ($s(\tau)$) and the probability of being infected ($p_s^{c1}(\tau)$). Based on the available data in our study, the mean of incubation period among vaccinated individuals was significantly shorter than that among unvaccinated individuals (P -value <0.001 , see Table R2 below). The impact of vaccination on the probability of being infected was not clear due to the limited data (see Table R3 below). Therefore, we can't directly evaluate the effects of vaccine based on the equation (15). We performed the sensitivity analysis in another way. Previous study showed that the inactive vaccines effectiveness against the Omicron infection was 6%-30% depending on the days after immunization (Jonathan J Lau *et al*, Nature Medicine, 2023). We assumed that the vaccine would reduce the basic reproduction number by 20% (from 10 in the main text to 8 in this sensitivity analysis) and performed the similar analysis. The results indicated that the super-spreading risk for the cospace-time interaction was still the lowest (Figure R1).

Figure R1. The sensitivity analysis for the super-spreading risk when the vaccination would reduce the basic reproduction number by 20%.

Ref:

Lau JJ, Cheng SMS, Leung K, Lee CK, Hachim A, Tsang LCH, Yam KWH, Chaothai S, Kwan KKH, Chai ZYH, Lo THK, Mori M, Wu C, Valkenburg SA, Amarasinghe GK, Lau EHY, Hui DSC, Leung GM, Peiris M, Wu JT. Real-world COVID-19 vaccine effectiveness against the Omicron BA.2 variant in a SARS-CoV-2 infection-naive population. *Nat Med.* 2023 Feb;29(2):348-357.

1) Here are the details about the impact of vaccines on the mean of incubation period:

Based on the COVID-19 cases with known exposure time, we estimated the mean of incubation period for the Omicron variant. A Weibull distribution show the best fit with the data compared to gamma and lognormal distributions using on a likelihood-based method (Table R2). Based on the data, the mean of incubation period is estimated to be 4.2 days (SD: 1.8 days) for unvaccinated cases and 3.3 days (SD: 2.0 days) for vaccinated cases (i.e., 2 or 3 doses). By comparison, we found that the mean of incubation period among vaccinated individuals is significantly shorter than that among unvaccinated individuals (P -value <0.001), indicating the effect of immunity induced by COVID-19 vaccines.

Table R2. Summary of incubation period estimates for COVID-19 cases infected by the Omicron variant by vaccination status in Beijing, China during the Spring of 2022.

Number of doses	Distribution		Estimate (95%CI)		LOO IC ^s
	Type of distribution	Sample size	Mean (days)	SD (days)	
	Weibull		4.2 (3.5, 5.0)	1.8 (1.5, 2.5)	102.4
	Gamma		7.8 (2.6, 20.4)	3.3 (1.4, 6.8)	102.3

0 doses	Lognormal	n = 25	4.3 (3.5, 5.4)	2.2 (1.5, 3.8)	103.8
2 doses	Weibull	n = 64	3.2 (2.8, 3.6)	1.7 (1.4, 2.1)	248.4
	Gamma		2.1 (1.0, 3.9)	1.3 (0.8, 2.1)	269.2
	Lognormal		4.9 (3.5, 7.3)	7.7 (4.6, 15.5)	346.1
3 doses	Weibull	n = 127	3.4 (3.1, 3.8)	2.1 (1.9, 2.5)	541.0
	Gamma		0.9 (0.5, 1.4)	0.7 (0.5, 0.9)	564.9
	Lognormal		7.7 (5.5, 15.8)	22.0 (12.5, 45.8)	714.4
2 or 3 doses	Weibull	n = 191	3.3 (3.1, 3.6)	2.0 (1.8, 2.3)	786.9
	Gamma		1.1 (0.7, 1.6)	0.8 (0.6, 1.1)	822.2
	Lognormal		6.6 (5.1, 8.8)	15.4 (10.2, 25.3)	1038.3

[§]LOO IC indicates the goodness-of-fit, where lower values indicate a better fit and differences larger than two are statistically relevant

2) Here are the details about the impact of vaccines on the infectiousness:

In general, the Ct value, reflecting the viral load, can be used as an indicator of the infectiousness. Therefore, we performed the analysis based on Ct value. Although multiple PCR tests were conducted for each case, only the Ct value and the corresponding sampling date of the first positive PCR test were documented in our dataset. The Ct values were aligned with the time of disease progression, using the symptom onset date as a reference. It is important to emphasize that only cases with symptoms were considered, and there was only one Ct value recorded for each case. Figure R2 illustrates the Ct values stratified by vaccination status. It's worth noting that Ct values for unvaccinated cases were deemed less reliable due to small sample sizes, particularly in the late phase of infections (the cutoff of 10 samples were used in this analysis). No significant differences at 5 time points were observed between unvaccinated and vaccinated cases (as shown in Table R3). Note that the Ct value for the entire disease progress was not evaluated due to the limited data.

Table R3. Comparison of Ct values between different vaccination status. P-value was obtained from t-test.

Vaccination Status	Statistics	Time since symptom onset (days)				
		-1	0	1	2	3
unvaccinated (0 doses)	sample size	24	75	40	17	10
vaccinated (2 or 3 doses)	sample size	149	454	357	152	71
	p-value of t-test	0.9579	0.7068	0.1744	0.6386	0.2822

Figure R2. The Ct values stratified by vaccination status. The sample size for unvaccinated cases was 176. Sample size for those receiving two or three doses was 1,291. Only cases with symptoms were considered here. The time interval between the date of the first positive PCR test and the date of symptom onset was calculated. The Ct values were aligned based on this time interval.

3) If the unknown settings are also classified as the cospace-time interactions, what will be the estimates of the transmission contribution and the super-spreading risk of the cospace-time interactions without control measures?

Response: Thank you for the comment. We merged these contacts from these types of location and performed the analysis again. The transmission contribution is 16.46% and the super-spreading risk of the cospace-time interactions is 6.06%, still the lowest among different locations (Figure R3).

Figure R3. The contributions to transmission and the super spreading event risk after classifying the unknown settings as the cospace-time interactions.

4) In lines 87-92, the authors presented three reasons of why that the age was not considered. In specific, “(3) the age distribution of infections closely resembled the overall age demographics in Beijing”, however, as presented in Table S5, the fraction of individuals with age larger than 60 in the sample is less than half of the population of Beijing for both the age distributions of the infections with identified contacts and contacts. The sample should be adjusted according to the age demographics in Beijing.

Response: Thank you for this helpful suggestion. To align the age distribution in our dataset with the age demographics in Beijing, we subsampled the cases based on the age demographics in Beijing for 100 times and performed the same analysis. The estimated means and variances of contact number per day of tracing under different locations were similar to previous analysis (Table S6). The transmission contribution is 10.97% (95%CI: 9.81%-12.40%) and the super-spreading risk of the cospace-time interactions is 3.79% (95%CI: 2.52%-5.32%). These results were consistent with those in the previous analysis. We also updated Figure 3 based on the adjusted contacts in the revised manuscript.

Reviewer #2 (Remarks to the Author):

Very useful work documenting the breakdown of infection based on location and number of contacts. Several comments:

We thank the reviewer for the positive feedbacks and the insightful comments and suggestions. We performed several sensitivity analyses to illustrate the robustness of our conclusions and updated the manuscript and supplementary materials accordingly. Please check the response below.

1. "Cospace time interaction" isn't a typical phrase. The authors have provided a description of the differences between cospace time interaction and community interactions. Furthermore cospace time interaction seems to be a mix of contacts in public settings and enclosed settings. Given that it forms more than 35% of the transmission pairs, perhaps the authors could provide a more granular breakdown of the public Vs enclosed space interactions.

This is important for policy making as the type of settings needs to be articulated to determine the type of public health control measures (e.g. physical distancing if in public Vs improve ventilation if in enclosed space)

Response: We appreciate the reviewer's comment. We fully agree with the reviewer that a more detailed breakdown of cospace-time interactions would offer valuable insights for policymaking. However, the detailed the public and enclosed space interactions are not available in this dataset. Our study revealed that 37.99% of cases occurred through cospace-time interactions with control measures. The proportion of transmission pairs in public and enclosed space interactions separately would be smaller than that for cospace-time interaction. Based on our updated analysis, the cospace-time interactions contributed to only 10.97% (95%CI: 9.81%-12.40%) of transmission and 3.79% (95%CI: 2.52%-5.32%) of super-spreading (SSE) risk without control measures. It is reasonable to expect that transmission contribution and SSE risk for public and enclosed space interactions would be lower than those for cospace-time interactions. Strategies such as physical distancing in public space and improving ventilation in enclosed space can further reduce the transmission contribution and SSE risk. We have included these limitations in the Discussion section (Line 196-197 of Page 5).

2. For the probability of infection of 0.87% in cospace time interaction (line 107) is this before or after control measures. Would be nice to have list of these probabilities before and after control measures.

Response: Thank you for the suggestions. We apologize for any lack of clarity in the previous version of the manuscript. Here we would like to characterize the probability of being infected over

the disease progression (Figure S6 in the revised Supplementary Materials), which reflects the biological nature of SARS-CoV-2. This probability is a function of the time since infection and remains independent of control measures. However, it's worth noting that control measures can mitigate infection risk by reducing contacts with infected individuals. We have revised these sentences in Line 130-133 of Page 4 in the revised main text.

3. line 65, how was contact tracing done? It's hard to understand 58 contacts were made over how many days.

Response: Thank you for the helpful suggestion. We are sorry for the lack of clarity. The contact tracing window varied among the primary cases based on Beijing CDC criteria (see below). In our study, we defined this window as the period from the first to the last day with recorded contacts. For days without recorded contacts, we couldn't differentiate whether tracing was conducted with no contacts found or if the day fell outside the tracing window defined by Beijing CDC. Our defined window implies normal activity during this period. The distribution of tracing window in our dataset was shown in Figure S2, with median of 3 days (95%CI: 1-11). We counted the contacts identified during the tracing window for each primary case and subsequently calculated the median number of contacts per case, providing an overview of the outbreak. We added the details in the revised Supplementary Materials (Line 61-85 of Page 2-3).

The contact tracing window criteria employed by Beijing CDC were as follows: (1) Typically, the tracing window spans from 4 days prior to symptom onset or the date of positive sample collection to the day before isolation or quarantine begins; (2) For the primary cases with an unclear source of exposure, the tracing window may be extended based on the risk assessment by Beijing CDC experts, ranging from >4 days (up to 14 days) prior to symptoms onset or the date of positive sample collection to the day before isolation or quarantine begins; (3) For the primary cases with a clearly identified source of exposure and prompt isolation, the tracing window may be shortened to minimize disruption to citizens' daily lives.

4. Line 80 do the authors mean "effect" rather than "impact"? How does sample bias result in more cospace time interactions, and hence transmission pairs being detected? More elaboration might be needed and to put the outbreak control policy into context

Response: Thank you for the helpful comment. We are sorry for the lack of clarity. The control measures can indeed influence the observed proportions of transmissions in various locations. For example, if all the workplaces are closed, transmissions in that setting would cease to occur. Consequently, the collected data would lack information regarding workplace transmission, leading to an absence of observed transmissions in that context. In another words, the control measures can alter the weighting of samples across different settings, resulting in deviations from the natural progression of infectious disease. We have revised this part in Line 94-108 of Page 3-4.

5 line103, most contagious 1.25 days since infection. How does this translate to days pre symptoms onset? For outbreak control, time of infection is seldom observed hence easier to conduct ops based on time of symptoms

Response: Thank you for the insightful comment. The most contagious time, calculated at 1.25 days since infection, is determined at the population level. This value is derived by weighting the fraction of asymptomatic cases and the cumulative distribution function of the incubation period over the disease progression. The incubation period was fitted to Weibull distribution in our study (Table S4 and Figure S1 in the Supplementary Materials). According to the fitted Weibull distribution, the mean of incubation period is 3.46 days. We also estimated that the probability of an incubation period longer than 1.25 days since infection is 87.70%. This indicated that for symptomatic cases, most of them would have symptoms after 1.25 days since infection. However, the pre-symptomatic individuals can still contribute to SARS-CoV-2 transmission. To further explore the control measures based on symptom onset, we performed the sensitivity analysis by isolating the case after the symptom onset. The estimated transmission contribution and super spreading event (SSE) risk for cospace-time interactions were 11.09% and 2.84% (Figure R4), still the lowest among the all locations studies. These results indicated that the isolating the cases after the symptoms onset might not be very effective for Omicron variant.

Figure R4. The contributions to transmission and the super spreading event risk after isolating the cases after the symptom onset.

6 line 74, authors excluded contacts associated with infection whose tracking duration covered only one day. Why was this exclusion criteria applied and how many contacts were removed in the process.

Response: Thank you for the helpful suggestions. We apologize that we didn't make it clear. We added a flowchart for the quality control (see Figure S3 in Supplementary Materials). The contact tracing was performed in a way of primary case-ego. Here we would like to remove the cases whose

tracking duration covered only one day. By this step, 283 cases and corresponding 6,157 contacts were filtered. To get the distribution of contact number per day, we need to collect samples. We defined the average of contact number per day of tracing for each case as a sample. As the contact number for a single day may not be reliable, we selected the cases with more than one day tracing duration and calculated the averages as samples. Another reason is that the cases can be assumed to be independent (they are different individuals). We performed sensitivity analysis by including the 283 cases with one day of contacts observed. The results indicated that the means of contact number per day by including 283 cases are similar with those in main study (Table R4). The transmission contribution and the SSE risk under cospace-time interaction were still estimated to be the lowest (Figure R5).

Table R4. The estimated means and variances of contact number per day, contribution of transmission and SSE risk when performing sensitivity analysis by including 283 cases whose tracking duration covered only one day.

Location	The estimated contact number per day		The contribution of transmission	The SSE risk
	Mean	Variance		
Dwelling	2.92	6.25	40.19%	8.69%
Workplace	13.73	332.64	18.10%	6.30%
Cospace-time interaction	27.92	1092.49	10.88%	3.98%
Community settings	25.15	1102.38	14.50%	5.73%
Unknown settings	3.49	11.92	16.33%	4.53%

Figure R5. The contributions to transmission and the super spreading event risk when performing sensitivity analysis by including 283 cases whose tracking duration covered only one day.

7. Code could be made available with mockup data to trial run.

Response: Thank you for the helpful suggestions. We added the mockup data to the Github. Please check it with this link (<https://github.com/NinaAvalale/contact-tracing-main>).

Reviewer #3 (Remarks to the Author):

Review of “Revisiting the effectiveness of digital contact tracing for cospace-time interactions” by Wang et al for Nature Communications

Summary

The authors report on the analysis of a digital contact tracing (DCT) dataset from 1121 COVID-19 index cases (with day of symptom onset) and 166,456 contacts, some of which were found to be infected. From these data, they estimate the incubation period, the generation interval, the distributions of the daily contact rates among five settings, and the probabilities of transmission per contact in each setting. They calculate the contributions of each setting to SARS-CoV-2 transmission, and the probabilities of superspreading events occurring in the five settings. These findings are interpreted in the light of the effectiveness of DCT, by (implicitly) assuming that DCT would be especially useful to prevent superspreading events in the setting of cospace-time interactions, where classic contact tracing cannot be carried out. They conclude that a minor part of all transmission occurs in this setting, and that superspreading events are expected to be rare. Yet, they recommend further study to balance between the benefits and costs of DCT.

We thank the reviewer for the constructive comments and suggestions. We have incorporated all of them in the revised manuscript. In addition, we updated the details of the methods in the Supplementary Materials. Please check the response below.

General comments

The data and analysis are described in the Supplement. The study describes a unique and valuable dataset that can be used to estimate the contact rates and transmission probabilities for SARS-CoV-2 in different settings. Close reading of the Supplement, and looking at data and results left me with many questions, laid out in detail below. Because of all these questions I cannot judge the validity of methods and results at this moment. However, the dataset itself and the general idea about the data analysis is promising.

Apart from not understanding the technical details in the Supplement, I have difficulty in following the argument from results to conclusions about the effectiveness of DCT. The title and introduction suggest that the analysis assesses the effectiveness of DCT, but that is not really the case. In fact, the statistical results about contact rates and transmission probabilities are used to discuss the consequence for DCT, involving underlying assumptions about conditions under which DCT would be effective. I think it is interesting to discuss the results in this light, but the underlying assumptions should be made clearer. From what I read between the lines, I think that these implicit assumptions are that DCT is mainly effective in transmission in cospace-time interactions and of most use to prevent superspreading events. In addition to explicitly stating the assumptions/conditions, I would like to understand why they are made.

Response: We thank the reviewer for this critical comment. In this revised manuscript, we have changed them and the title is "Estimation of transmission contribution and the super-spreading risk for cospace-time interactions based on digital contact tracing" now. The underlying assumption is that digital contact tracing is empirically effective in preventing transmission and super-spreading events. This assumption is grounded in observations since the outbreaks in Beijing (and many places around the world) were controlled. To make it clear, we have revised the introduction in Line 60-72 of Page 3.

Detailed comments

A first comment is about the result that the distribution of transmission events by location (figure 1) does not match the distribution of the contributions to transmission by location (figure 3). Intuitively I would expect those to be the same: when 38% of transmission events occur in cospace-time-interaction, why is the contribution of cospace-time-interaction to transmission not just 38%. Also, when I read the model equations in the Supplement where I see that you assume that the infectivity profiles ($\omega(\tau)$, $\beta(\tau)$, $p(\tau)$) are identical for all locations apart from scaling constants, so I would expect this even more. The only part where the locations may differ is in the contributions of asymptomatic, presymptomatic and symptomatic transmission, but I cannot follow that part of the analysis. Anyway, I would like to see an intuitive explanation of why these distributions by location in figures 1 and 3 are not identical.

Response: Thank you for the insightful comment. The infectivity profiles [$\omega(\tau)$, $s(\tau)$] and the transmission category (i.e., asymptomatic, pre-symptomatic or symptomatic) can be treated as the biological nature of SARS-CoV-2 and should be independent of the locations. $\beta(\tau)$ is the product of contacts and the probability of being infected. Therefore, the parts where the location may differ are the contacts (c_i) and the probability of being infected [$(p_j^{ci}(\tau)$, reflected by scaling constants] in locations, which can represent the characteristics of human behavior and the environment specific to each location. Intuitively, in Figure 1, we indeed observed that approximately 38% of cases occurred through cospace-time interactions. However, it's crucial to note that this figure was obtained during the period of implementation of the Dynamic Zero COVID policy in Beijing. Control measures can influence the observed proportions of transmissions in different locations. For instance, if all workplaces are closed, we would expect to observe no transmissions in workplaces. In another words, the control measures can alter the weighting of samples across different settings, resulting in deviations from the natural progression of infectious disease. In our study, our aim is to quantify the transmission contributions of different locations under conditions where no control measures are implemented, as depicted in Figure 3. Therefore, it's expected that the numbers in Figure 1 and Figure 3 would differ.

Technically, the contact number per day for all locations would vary with and without control measures throughout the entire disease progression. In Figure 3, we used the normal contact number per day, and the corresponding cases were counted in the model. However, in Figure 1, the contacts were unobserved due to the control measures, resulting in corresponding cases not being observed.

Suppl, line 42: “locations were extracted”, but for which time window before detection of an infected individual?

Response: Thank you for the helpful suggestion. We are sorry that we didn't make it clear. The tracing window criteria employed by Beijing CDC were as follows: (1) Typically, the tracing window spans from 4 days prior to symptom onset or the date of positive sample collection to the day before isolation or quarantine begins; (2) For the primary cases with an unclear source of exposure, the tracing window may be extended based on the risk assessment by Beijing CDC experts, ranging from >4 days (up to 14 days) prior to symptoms onset or the date of positive sample collection to the day before isolation or quarantine begins; (3) For the primary cases with a clearly identified source of exposure and prompt isolation, the tracing window may be shortened to minimize disruption to citizens' daily lives. Both traditional and digital contact tracing were used to identify the locations and the corresponding contacts.

The contact tracing window varied among the primary cases based on Beijing CDC criteria. In our study, we defined this window as the period from the first to the last day with recorded contacts. For days without recorded contacts, we couldn't differentiate whether tracing was conducted with no contacts found or if the day fell outside the tracing window defined by Beijing CDC. Our defined window implies normal activity during this period. The distribution of tracing window in our dataset was shown in Figure S2, with median of 3 days (95%CI: 1-11). We added these details in Supplementary Materials (Line 60-85 of Page 2-3).

Suppl, line 47, 48, 54, 55: “de-identification”, what is that? Anonymized?

Response: Thank you. We are sorry for the lack of clarity. We would like to clarify that the personal information was removed from the dataset due to privacy concerns when it was available to us. We have revised these (Line 39-40 of Page 2, 91 and 94 of Page 3 in the Supplementary Materials).

Suppl, line 50: “monitoring period”, how long was that?

Response: Thank you for the comment. The contacts would undergo 10 days of centralized isolation followed by 7 days of home isolation, during which regular nucleic acid testing and antigen self-testing would be conducted according to the specified guidelines. We added these details in the Supplementary Materials (Line 96-98 of Page 3).

Suppl, lines 45 and 53 mention 1555 infections and 2230 infections, what is the difference between these numbers? Maybe the 1555 are the infections whose contacts were traced, but the description in lines 53-56 explicitly mention the structure of infector-infectee pairs which suggests that it is about traced individuals. I suggest you write one piece of text that links the information in lines 45-

56, and maybe also the steps in the quality control so that all numbers are explained in a single line of explanation.

Response: Thank you for the helpful suggestions. Yes, there are 1,555 cases whose contacts were traced. For the rest of cases (675=2230-1555), there were no recorded contacts for them (Line 89-90 of Page 3 in the Supplementary Materials). For the lines 53-56, we are sorry for the lack of clarity. Despite the infector-infectee pairs structure, there were missing values in this dataset, especially for the infectors. We revised 'Quality control for the contacts' section to improve the readability (Page 3 in the Supplementary Materials). We also added a flowchart plot for the data processing (Figure S3).

Suppl, lines 45-50: by contacts you mean contacts found by traditional contact tracing only?

Response: Thank you for the suggestion. No, the contacts identified by traditional and digital contact tracing were combined. We revised these sentences to make it clear in the revised supplementary materials in Line 87-88 of Page 3.

Supple, line 47: "contact mode", does that refer to the five locations in the analysis? "Entertainment" seems to be something else.

Response: Thanks for the comments. The contact mode contained the detailed information for the contact, including entertainment, work, living together, caregiving, sharing an elevator, ride-sharing, dining together, same apartment building and others. They were merged into these five locations based on Table S1.

Suppl, lines 73-77: "we excluded ... covered only one day" seems to result in exclusion of 431 infections (from 1552 to 1121 infections), that seems a lot. I don't understand why one-day observations are unreliable, except when these do not cover complete days. In fact, I think that the analyses is better done on daily numbers of contacts anyway (see below my comment on lines 98-106)

Response: Thank you for the helpful suggestions. We apologize that we didn't make it clear. We added a flowchart for the quality control (see Figure S3 in Supplementary Materials). The contact tracing was performed in a way of primary case-ego. Here we would like to remove the primary cases whose tracking duration covered only one day. By this step, 283 cases and corresponding 6,157 contacts were filtered. To get the distribution of contact number per day of tracing, we need to collect samples. There are two ways to define a sample. The first one is the average of contact number per day for each primary case. As the contact number for a single day may not be reliable, we selected the cases with more than one day tracing duration and calculated the average as samples.

The second one is the contact number per day for all primary cases during tracing duration (as you suggested). Note that the samples defined by this way contains multiple contact numbers from the same cases. We chose the first one because the cases can be assumed to be independent (they are different individuals). The contact number across multiple days for the same case may be highly correlated. This type of samples may introduce the bias for the distribution estimation of contact number per day. We performed sensitivity analysis using the second definition. The results indicated that the means of contact number per day in second definition are similar with those in the main text (Table R5). The transmission contribution and the SSE risk under cospace-time interaction are still estimated to be the lowest (Table R5 and Figure R6).

Table R5. The estimated means and variances of contact number per day, contributions of transmission and SSE risk when using contact number per day for all cases during tracing duration (i.e., the second definition, including the 283 cases with one day of contacts observed). Note that the samples defined by this way contains multiple contact numbers from the same cases.

Location	The estimated contact number per day		The contribution of transmission	The SSE risk
	Mean	Variance		
Dwelling	2.71	10.57	37.88%	11.11%
Workplace	13.05	588.34	17.47%	10.43%
Cospace-time interaction	29.78	2613.88	11.82%	7.90%
Community settings	20.69	1238.41	12.12%	8.19%
Unknown settings	4.36	49.47	20.71%	9.70%

Figure R6. The transmission contribution and the SSE risk using the second definition. (A) transmission contribution under different locations. (B) The SSE risk under different locations.

Suppl, line 83: “1495 transmission pairs”, does the difference between 1552 and 1495 mean that for 57 infections no infector could be found, or that there were multiple candidate infectors from

which it is impossible to select the right one? More general: were all the 1495 infectees in the pairs unambiguously linked to one infector? I can hardly imagine that that is the case, especially not with transmissions in dwellings where I would expect clusters of cases.

Response: We are sorry for the lack of clarity. The discrepancy between 1,552 and 1,495 does not imply that there were 57 cases without identifiable infectors. The 1,552 were the primary cases with recorded contacts. The 1,495 transmission pairs involved 451 infectors and 1,495 infectees. It's important to note that one infector may be linked to multiple infectees. The total number of cases is 1,745, as determined by unique case's IDs (combining infectors and infectees). We added this clarification in Line 157-158 of Page 4 in the Supplementary Materials.

The 1,495 recorded transmission pairs were thoroughly investigated by experts from Beijing CDC. They utilized digital technology to verify the activity trajectory of the infected person 14 days before the onset of symptoms/positive sample collection until the implementation of isolation. Multiple methods, such as contact tracing, whole-genome sequencing, activity trajectory comparisons, the nucleic acid testing results in the environments, were used to identify the transmission chain. Additional information, such as the onset time, incubation period and duration of contact, were used to enhance the precision of the source of case identification. Unfortunately, the specifics of the transmission chain investigation were not available to us. Furthermore, 26 rounds of population-level PCR testing were implemented during this outbreak, which helped the infection status surveillance among Beijing residents. Indeed, there was clusters of cases in this outbreak.

Suppl, lines 88-95 and Table S4, estimation of incubation period and generation interval distributions: model selection is done and the Weibull models turn out to fit best. However, the estimated means and variances of the three distributions (Weibull, gamma, lognormal) are very different, which suggests that something goes wrong in the estimation procedure. You would expect that the estimated means and variances are similar to the means and variances directly calculated from the data. Can you provide these as well and/or plot the data as histograms in Figure S1?

Response: Thank you for the helpful suggestion. For generation interval, the mean and SD directly calculated from the data were 3.00 and 1.83, respectively. For incubation period, they are 3.53 and 1.83. The estimated mean and variance based on Weibull distribution are similar to the means and variances directly calculated from the data. We add these to Table S4. We also plot the histograms and found that Weibull distribution fit well (Figure R7).

Figure R7. The histograms for generation interval (A) and incubation time (B).

Suppl, lines 98-106: The mean number of contacts is calculated per case, and then the means are fitted to distributions. Are these means of just a few days of observation per case around the time of possible onward transmission, or are these based on much longer observation periods starting long before infection? That would be relevant, also in relation to the 431 excluded infections with one day of observation.

Response: Thank you for the comment. As the response to above comments (Suppl, line 42) showed, we defined this window as the period from the first to the last day with recorded contacts in our study. For days without recorded contacts, we couldn't differentiate whether tracing was conducted with no contacts found or if the day fell outside the tracing window defined by Beijing CDC. Our defined window implies normal activity during this period. With 26 rounds of population-level PCR testing implemented during this outbreak, the cases should be identified timely. Based on the distribution of tracing window in our dataset (Figure S2 in the Supplementary Materials), the majority of the means could be a few days of observation per case around the time of possible onward transmission.

Suppl, lines 98-106: The mean number of contacts is calculated per case, and then the means are fitted to distributions. These distributions are used to calculate the probability of a superspreading event (SSE), as the probability of exceeding the Poisson-99% percentile. However, SSEs are single events on a single day, so it is more relevant to know the distribution of contact numbers per day, not the distribution of means. And if you would estimate the distribution of contact numbers per day, then why not include the 431 infections with one day of contacts observed?

Response: Thank you for this comment. As we described in the comment of Suppl, lines 73-77, one concern is the dependence of contact numbers over the tracing window for a primary case. In addition, we believed that the one-day observations may not be reliable for that case (i.e., only one sample). We performed sensitivity analysis by estimating the distribution of contact number per day. The transmission contribution and the SSE risk under cospace-time interaction are still estimated to be the lowest (Table R5 and Figure R6). Note that for each location, the cases with one-day observation and zero contacts in the corresponding location were excluded in this sensitivity analysis since these zeros may not be the true value given the Dynamic zero COVID policy implemented at that time. However, the zero contacts for the cases with more than one-day observations in the corresponding location were kept. Our rationale is that the non-zero contacts should be more reliable than the zero-contacts and the long tracing window should be more reliable than the short tracing window, if aiming to get the contact patterns and minimizing the impact of control measures on these patterns.

Suppl, lines 98-106: A suggestion. In the mathematical model, there is a factor for the transmissibility of asymptomatic, pre-symptomatic and symptomatic individuals. Transmissibility may be related to viral load, but also to contact frequencies which may decrease if people get ill. If I understand correctly, you have symptom onsets in your data and you have asymptomatic infections. Is it possible to estimate contact rates for these three categories separately, especially symptomatic vs not-symptomatic?

Response: Thank you for the insightful suggestion. In this analysis, we focused on comparing contacts between symptomatic and pre-symptomatic cases, as contacts should be similar for pre-symptomatic and asymptomatic cases. Our results indicated that the contact patterns before and after symptom onset were similar across all locations (Table R6). To understand the reason behind this, we checked the tracing window for the symptomatic cases. With 159 symptomatic cases with observed contacts after symptom onset, only 44.03% had a contact tracing window longer than 1 day. Therefore, we speculated that the short tracing window after symptom onset could lead to unreliable estimation in our dataset. Generally, individuals may not change their behavior immediately after symptom onset. Given the rapid isolation of SARS-CoV-2 cases in Beijing, analyzing contact patterns after symptom onset was not feasible based on our data. Considering the decreased contacts if people get ill, we performed the sensitivity analysis by isolating the case after the onset of symptoms. The estimated transmission contribution and super spreading event (SSE) risk for cospace-time interactions were 11.09% and 2.84% (Figure R4 in the response of Reviewer #2), still the lowest among the all locations studies.

Table R6. The contact number per day comparison between before symptom onset and after symptom onset.

Type of locations	Types of contacts	Sample size	Median	P -value of the Wilcoxon test
Dwelling	before symptom onset	n = 203	2.00	0.1342

	after symptom onset	n = 38	2.00	
Workplace	before symptom onset	n = 367	3.50	0.1477
	after symptom onset	n = 34	4.50	
Cospace-time interactions	before symptom onset	n = 862	13.27	0.9717
	after symptom onset	n = 133	13.00	
Community settings	before symptom onset	n = 210	6.00	0.2511
	after symptom onset	n = 35	7.50	
Unknown settings	before symptom onset	n = 248	1.71	0.9750
	after symptom onset	n = 29	2.00	

Suppl, section 3: I can follow the mathematics of the model, but not how the model is used for inference.

1. From what I understand, the generation interval distribution $\omega(\tau)$ is the sum of the three $\beta(\tau)$ functions of the three infection categories, and the equation in line 135 tells me that $\omega(\tau)$ and $\beta(\tau)$ are different only because of x_p and $s(\tau)$ which makes that $\beta(\tau)$ gets a different weight in the time since infection. If you estimate p_a from the data in Table S2 (19.9% asymptomatic) and $\omega(\tau)$ and $s(\tau)$ are as estimated in Table S4 (please make that clear somewhere!), then the equation in line 135 has 3 unknowns: x_a , x_p , and $\beta_S(\tau)$. How are these estimated? Lines 189-191 only tells that you “solved $\beta_S(\tau)$ through fitting...”, but not how this was done exactly. Do you assumed x_a and x_p to be 1? And then, is $\beta_S(\tau)$ calculated numerically? Then better say that you solved $\beta_S(\tau)$ numerically from the equation instead of fitting, which suggests a statistical procedure.

Response: Thank you for the helpful comments. Yes, the $\omega(\tau)$ and $s(\tau)$ were estimated based on Table S2 and Table S4. p_a was set to be 21.8% based on data for primary cases, which was close to previous study (Yu W *et al*, J Med Virol, 2022). x_a and x_p was set to be 1 based on previous studies (Zhongqiu Wei *et al*, Influenza Other Respir Viruses, 2023; Xueting Qiu *et al*, Clin Microbiol Infect. 2021; Luca Ferretti *et al*, Science, 2020). Using the Ct value (Figure R2 in the response of Reviewer #1), we also found that there was no significant difference of Ct value between the days before symptom onset and the days after symptom onset (p-value: 0.64). This is another evidence to support $x_p=1$. Then we can solve $\beta_S(\tau)$ numerically from the equation. We recognized that the fitting was not a proper word and we have revised it. In addition, we also revised the estimations for the parameters to improve the readability (Line 361-397 of Page 7-8 in the Supplementary Materials). Note that x_a and x_p could be set to other values from mathematical view. However, we set them to be 1 based on the empirically epidemiological evidence, which is a very special scenario. Other values for them would not alter the conclusions about transmission contribution and SSE risk, because they depend on the contact number under different locations based on equation (15).

Ref:

1. Yu W, Guo Y, Zhang S, Kong Y, Shen Z, Zhang J. Proportion of asymptomatic infection and nonsevere disease caused by SARS-CoV-2 Omicron variant: A systematic review and analysis. *J Med Virol.* 2022 Dec;94(12):5790-5801.
2. Wei Z, Ma W, Wang Z, Li J, Fu X, Chang H, Qiu Y, Tian H, Zhu Y, Xia A, Wu Q, Liu G, Zhai X, Zhang X, Wang Y, Zeng M. Household transmission of SARS-CoV-2 during the Omicron wave in Shanghai, China: A case-ascertained study. *Influenza Other Respir Viruses.* 2023 Feb;17(2):e13097.
3. Qiu X, Nergiz AI, Maraolo AE, Bogoch II, Low N, Cevik M. The role of asymptomatic and pre-symptomatic infection in SARS-CoV-2 transmission-a living systematic review. *Clin Microbiol Infect.* 2021 Apr;27(4):511-519.
4. Ferretti L, Wymant C, Kendall M, Zhao L, Nurtay A, Abeler-Dörner L, Parker M, Bonsall D, Fraser C. Quantifying SARS-CoV-2 transmission suggests epidemic control with digital contact tracing. *Science.* 2020 May 8;368(6491):eabb6936.

2. The scaling factors y_{ci} are estimated from the proportions of infected contacts per scaling location (lines 191-194), from which I understand that you use the fractions at the bottom of Table S3 (please make that clear!).

Response: Thank you for the comment. Yes. We have revised it to improve the readability (Line 367-368 of Page 7 in the Supplementary Materials).

3. Then I understand that you use the c_i as the means of the estimated distributions as described in lines 98-106. I think that the means of these distributions should be proportional to the denominators of the probabilities at the bottom of Table S3, but maybe this is not exactly so because the distributions are fitted with means per infection whereas the denominators in Table S3 are the totals. If you would fit the distributions to daily observations (as I suggest), I think the means of the distributions should really become proportional to the denominators in Table S3, so that the product of y_{ci} and c_i will simply reflect the proportions of cases observed in the five settings (numerators at the bottom of Table S3, and Figure 1 main text).

Response: Thank you for the insightful comment. Based on the results in the previous version, we calculated the correlation between c_i and the denominators of the probabilities at the bottom of Table S3, which is 0.74. As suggested, we also estimated the contact number per day based on the daily observations. The correlation between the means and the denominators in Table S3 is 0.85. This indicated that the means seems to be proportional to the denominators in Table S3. However, the product of y_{ci} and c_i were 2.71 for dwelling, 1.25 for workplace, 0.85 for cospace-time interaction, 0.87 for community settings and 1.48 for unknown settings. The correlation between the product of y_{ci} and c_i and the numerators at the bottom of Table S3 is -0.53. The possible reason for this negative correlation is that the c_i represents the normal contact number per day. The product of y_{ci} and c_i represents the corresponding cases counted in the model. However, in

Figure 1, the contacts were unobserved due to the control measures, resulting in corresponding cases not being recorded or counted.

4. In lines 194-199, a large number M is introduced but I don't understand why and what the value would be. Then, equations 5 and 8 are used to solve $p_S^{c1}(\tau)$, but from what I understand from lines 174-177 and eq(8), all functions $p_j^{ci}(\tau)$ are only different by the constants y_{ci} and c_i , the product of which would be the relative contribution to transmission (equation 14). Thus, I don't understand what is meant by solving $p_S^{c1}(\tau)$.

Response: Thank you for the constructive comment. C_i represents the contact number per day of tracing, which was estimated in our study. c_i represents the contact number at time τ (i.e., the contact rate) and it belongs to the integral term. Here we assume that c_i remain constant over time. Therefore, we approximated the contact rate as C_i/M , where M is set to be 20. Indeed, there is no need to solve $p_S^{c1}(\tau)$ if only the relative contribution to transmission is focused. We solved $p_S^{c1}(\tau)$ for two purposes: (1) dynamic infectiousness under different locations (i.e., Figure S6) is also our interest, which can be treated as biological nature of SARS-CoV-2; (2) With solved $p_S^{c1}(\tau)$, we can evaluate the effects of control measures on the transmission contribution in the future study, especially for the time-dependent control measures. The c_i can reflect the human behavior in response to the control measures. We added these in Line 392-397 of Page 8 in the supplementary materials.

Regarding all points 1-4: maybe I misunderstood what was done, but then I would encourage the authors to explain the calculations/estimations in lines 189-199 in more detail.

Response: We appreciate the reviewer for these constructive comments. We have revised the section to improve the readability of our manuscript (Line 361-397 of Page 7-8 in the Supplementary Materials).

REVIEWER COMMENTS

Reviewer #1 (Remarks to the Author):

The authors have addressed all my questions and I'd like to see this work being published.

Reviewer #2 (Remarks to the Author):

The authors have addressed my questions accordingly.

Reviewer #3 (Remarks to the Author):

Review of revised manuscript "Estimation of transmission contribution and the super-spreading risk for cospace-time interactions based on digital contact tracing" by Wang et al for Nature Communications

General comments

I had two major comments with the initially submitted manuscript. One was about the message of the analyses, which is estimating the contributions of transmission settings (especially cospace-time interactions) to overall transmission of SARS-CoV-2. I think the authors addressed this comment well. The other was about my understanding of the actual analyses, as explained in the Supplement. The authors made some adjustments there, but have not resolved the problem. I still cannot follow how the data have been used to obtain the results.

One point I did not understand with the previous version was the difference between the contributions to transmission of the five locations in Figures 1 and 3. Now, the authors clearly explain why they are different and how they should be interpreted: Figure 1 represents the observed distribution, and Figure 3 a modelled distribution extrapolated to the situation without NPIs. However, the methods in the Supplement I cannot understand how these are obtained. First of all, nowhere is explained conceptually what information/data is used to make the translation from the observed contact rates and transmission probabilities (Tables S3 and S6) to how these will be in the counterfactual situation without control NPIs. Second, reading the mathematical details sentence by sentence, there are several points where I do not understand what is actually done. I will list some examples below, but I must add that there is a lot of redundancy in the first part of the mathematical model description which makes it sometimes difficult to read.

Detailed comments (line numbering across pages is strange)

Supplement lines 172-265: distributions of numbers of contacts per day are estimated. This is used to draw conclusions about superspreading events. As I suggested with the previous version, I think this should be done with daily contact numbers, not means of daily contact numbers. From the reply in the rebuttal letter I see that this has to do with repeated contacts during the tracing window. I realise the complication, but I do not understand how this is solved by averaging across the tracing window: then, repeated contacts also end up more than once, aren't they? Also, for superspreading events, I still think that the distribution of the number of contacts per day is most relevant.

Supplement, section 3 (lines 267 and below):

1. Describing what I read and understand: in eq (1), $\omega(\tau)$ is introduced as the generation interval distribution, but $\omega(\tau)$ is not used in any equation until it is estimated in line 364. That is a bit confusing, but for now I understand that the shape of $\beta(\tau)$ and the shape of $\omega(\tau)$ are the same.
2. Describing what I read and understand: In eq (3), $\beta(\tau)$ is split into an asymptomatic, pre-

symptomatic and symptomatic part. In eq(4), the assumption is made that the three all have the same shape, which is therefore equal to the shape of $\omega(\tau)$. Later (line 365), the values of x_a and x_p are assumed to be 1. As a result, eq (5) will be simplified to $\beta(\tau) = \beta_S(\tau)$.

3. Describing what I read and understand: In eq(6), $\beta_S(\tau)$ is defined as a sum of five contributions for the five locations. In eq(9) the contributions are split into a contact rate (independent of time) and a probability of transmission $p_S^{ci}(\tau)$ (dependent of time). In line 348, the assumption is made that the time-dependent probabilities between the settings are only different by a scaling factor. Therefore, $p_S^{ci}(\tau)$ all have the same shape as $\omega(\tau)$.

4. Describing what I read and understand: from all the above, all that remains is $R = \sum_i [c_i * y_{ci} * \int p_S^{c1}(\tau) d\tau]$. In words: the reproduction number is a sum of the contribution of the five settings i , which ONLY differ by their mean contact rate c_i and their mean probability of transmission per contact y_{ci} , relative to that probability in setting 1 (the integral).

5. Concluding from the above descriptions: I can understand how you can estimate c_i and y_{ci} from the numbers in Table S3. What I do not understand is how you can use these to calculate the contributions to R , in the absence of NPI. What data are available, and what values do change in the above model?

Line 361 "numerically solving $\beta_S(\tau)$ from the inferred generation interval distribution": according to the above observation #1, the shapes of $\beta_S(\tau)$ and $\omega(\tau)$ are identical, so what is there to solve? The values in lines 363-365 are not needed.

Line 365 " R_0 was set to 10": why would you need a value for relative contributions?

Lines 365-368 "Subsequently, we could determine the proportion of infected contacts...": which parameter in the model do you calculate here? And how do you calculate y_{ci} ?

Lines 368-369 "Furthermore, we derived the distribution for c_i " and further: the model is all about means and expectations, which is obvious given that the calculations are about the reproduction number. Therefore, I do not understand why there is any mention of distributions here. I also do not understand the difference between c_i and C_i (apart from a factor 20??). What is really puzzling is the phrase in line 386 "contact number at time τ ", which suggests that c_i is not a constant.

Lines 388-397: At this point I'm really lost, and I don't follow how the observed contributions are translated to contributions under the 'biological nature of SARS-CoV-2', i.e. without NPIs

Manuscript ID: NCOMMS-23-58901A

General Response:

We greatly appreciate the time and effort that the reviewers dedicated to providing feedback on our manuscript. Their insightful comments and suggestions have been invaluable in improving the quality of our paper. We have made slight revisions to our manuscript title, which is "Estimating the contribution to transmissions and the super-spreading risk for cospacetime interactions based on digital contact tracing". We have carefully incorporated all the suggestions made by the reviewers, particularly focusing on enhancing the readability of the Supplementary Materials. We have also standardized the significant figures throughout the main text. Those changes are highlighted in the manuscript.

Please find our responses below, in blue, for a point-by-point response to the reviewers' comments and concerns. All page and line numbers refer to the revised manuscript with tracked changes.

REVIEWER COMMENTS

Reviewer #1 (Remarks to the Author):

The authors have addressed all my questions and I'd like to see this work being published.

Response: We thank the reviewer for this positive feedback of our revised manuscript. We greatly appreciate the kind words.

Reviewer #2 (Remarks to the Author):

The authors have addressed my questions accordingly.

Response: We thank the reviewer for this positive evaluation of our revised manuscript. We greatly appreciate the kind words.

Reviewer #3 (Remarks to the Author):

Review of revised manuscript “Estimation of transmission contribution and the super-spreading risk for cospace-time interactions based on digital contact tracing” by Wang et al for Nature Communications

General comments

I had two major comments with the initially submitted manuscript. One was about the message of the analyses, which is estimating the contributions of transmission settings (especially cospace-time interactions) to overall transmission of SARS-CoV-2. I think the authors addressed this comment well. The other was about my understanding of the actual analyses, as explained in the Supplement. The authors made some adjustments there, but have not resolved the problem. I still cannot follow how the data have been used to obtain the results.

Response: We are grateful to the reviewer for the feedback to our revision, which have been extremely helpful in improving the readability of this work. We have addressed the comments below by providing additional details in the Supplementary Materials and including further clarifications throughout the manuscript. We extend our thanks to the reviewer once again.

One point I did not understand with the previous version was the difference between the contributions to transmission of the five locations in Figures 1 and 3. Now, the authors clearly explain why they are different and how they should be interpreted: Figure 1 represents the observed distribution, and Figure 3 a modelled distribution extrapolated to the situation without NPIs. However, from the methods in the Supplement I cannot understand how these are obtained. First of all, nowhere is explained conceptually what information/data is used to make the translation from the observed contact rates and transmission probabilities (Tables S3 and S6) to how these will be in the counterfactual situation without control NPIs. Second, reading the mathematical details sentence by sentence, there are several points where I do not understand what is actually done. I will list some examples below, but I must add that there is a lot of redundancy in the first part of the mathematical model description which makes it sometimes difficult to read.

Response: We thank the reviewer for this comment. We calculated the transmission contribution and SSE risk without NPIs based on the following information: (1) The samples for the contact number per day of tracing in Figure 2 exclude cases that were more likely influenced by NPIs (i.e., individuals who had zero contacts recorded in the corresponding location). The contact number per day of tracing in Figure 2 can be treated as the normal contact pattern (i.e., without NPIs); (2) The transmission probabilities under different locations (Figure S6) were our interest. We obtained them under $R_0=10$ (i.e., without NPIs) and the normal contact pattern. Thus, they can be treated as the biological nature of SARS-CoV-2. While these probabilities may depend on

control measures, assessing their dependence on control measures is challenging. To our knowledge, the only high-quality work on this subject is Sender et al 2022 (<https://elifesciences.org/articles/79134>), which focuses on the original SARS-CoV-2 strain rather than variants. Considering the large amount of traced contacts, the effects of control measures on these probabilities should be limited in our study.

Based on the transmission contribution, we obtained the basic reproduction number and the distribution of secondary cases for each location, allowing us to estimate the SSE risk. Thus, the transmission contribution and SSE risk from this study can be treated as the situation without NPIs. We added these explanations into Supplementary Materials (Line 318-329 of Page 8 in the revised Supplementary Materials). For the mathematical part, we have revised it to improve the readability in the revised Supplementary Materials.

Regarding why the contact number per day of tracing in Figure 2 can be treated as the normal contact pattern, we provided the following justifications:

In total, 2,230 cases and 220,878 contacts were identified, with median number of contacts per case of 58.00 (95% CI: 1.00-787.13). This demonstrates a very intensive contact tracing. The uniqueness of our dataset is that the contacts who tested negative for SARS-CoV-2 were included in this dataset. This allows us to capture the social contact patterns (Line 118-119 of Page 4 in the revised Main Text). In addition, we defined this tracing window as the period from the first to the last day with recorded contacts. This definition implies normal activity before the quarantine or isolation of cases (Line 76-77 of Page 3 in the revised Supplementary Materials). The samples for the contact number per day of tracing in Figure 2 exclude cases that were more likely influenced by NPIs (i.e., individuals who had zero contacts recorded in the corresponding location). Therefore, the contact number per day of tracing in Figure 2 can be treated as the normal contact pattern (i.e., without NPIs). During the outbreak period, the contact pattern may be impacted by the implemented dynamic zero COVID policy. We adjusted the contact patterns based on intra-city mobility data from Beijing and the conclusions did not vary significantly (Line 154-158 of Page 4 in the revised Main Text and Figure S7-S8 in revised Supplementary Materials).

Detailed comments (line numbering across pages is strange)

Response: We apologize for the inconvenience. To ensure the equations are displayed correctly, we have submitted the PDF files instead of the Word files. Thank you for your understanding.

Supplement lines 172-265: distributions of numbers of contacts per day are estimated. This is used to draw conclusions about superspreading events. As I suggested with the previous version, I think this should be done with daily contact numbers, not means of daily contact numbers. From the reply in the rebuttal letter I see that this has to do with repeated contacts during the tracing window. I realise the complication, but I do not understand how this is solved by averaging across

the tracing window: then, repeated contacts also end up more than once, aren't they? Also, for superspreading events, I still think that the distribution of the number of contacts per day is most relevant.

Response: We appreciate the reviewer's comment. We realized that we misunderstood your previous comments, and we apologize for any confusion caused.

In general, the distribution of secondary cases is characterized by a negative binomial distribution with a mean of the basic reproduction number (R_0) and a dispersion parameter (Lloyd-Smith *et al*, Nature, 2005). The SSE is defined as the secondary cases of a case exceeding the 99th percentile $Z(R_0)$ of a Poisson distribution with R_0 as the mean (Lloyd-Smith *et al*, Nature, 2005). In our study, we assumed that the contact number per day of tracing followed a Gamma distribution. Therefore, the distribution of secondary cases for each location also follows a Gamma distribution with a mean of R_{loc} , where R_{loc} represents the basic reproduction number for a location. In other words, we focused on the superspreading events caused by the variation in the contact pattern. While heterogeneity in infectiousness among different cases may also contribute to superspreading, we ignored it due to the lack of corresponding data. We used the distribution of contact number per day to derive the distribution of secondary cases and then calculated SSE risk for each location, not means of daily contact numbers. We have added the specific distribution of secondary cases in the revised Supplementary Materials (Line 332-336 of Page 8).

The next question is how to derive the distribution of contact number per day of tracing. To obtain the distribution of contact number per day of tracing, we need to collect samples and fit the distribution based on the samples. There are two ways to define a sample. The first one is the average of contact number per day for each primary case. Note that the averages of contact number across the tracing window were samples. The second one is the contact number per day for all primary cases during tracing duration (as you suggested in previous comments). In the previous response letter, we have shown that the ways to fit the distribution of contact number per day of tracing do not change the conclusions. We would like to characterize the daily contact numbers during the whole disease progression without NPIs. However, this type of data was not available. The repeated exposures were not captured in our study and may underestimate the contact number per day, especially for dwelling. However, this wouldn't alter our conclusions, as repeated exposures in dwelling would increase their transmission contribution. We have added this as a limitation (Line 197-199 of Page 5 in the revised main manuscript).

Ref:

Lloyd-Smith, J. O., Schreiber, S. J., Kopp, P. E. & Getz, W. M. Superspreading and the effect of individual variation on disease emergence. Nature 438, 355–359 (2005).

Supplement, section 3 (lines 267 and below):

1. Describing what I read and understand: in eq (1), $\omega(\tau)$ is introduced as the generation interval distribution, but $\omega(\tau)$ is not used in any equation until it is estimated in line 364. That is a bit confusing, but for now I understand that the shape of $\beta(\tau)$ and the shape of $\omega(\tau)$ are the same.

Response: We thank the reviewer for this comment. Yes, the shape of $\beta(\tau)$ and the shape of $\omega(\tau)$ are the same based on eq (1). $\omega(\tau)$ was utilized when we numerically solved $\beta_s(\tau)$. When $x_a = x_p = 1$ (a very special scenario), the shape of $\beta_s(\tau)$ is same as the shape of $\beta(\tau)$. Therefore, the shape of $\beta_s(\tau)$ is same with the shape of $\omega(\tau)$. If other values were set for x_a and x_p , $\beta_s(\tau)$ would have to be numerically solved. This modelling framework can be applied to other infectious diseases with modification of epidemiological parameters. Therefore, we described this modeling part for generality in this revised Supplementary Materials (Line 247-255 of Page 7 in revised Supplementary Materials).

2. Describing what I read and understand: In eq (3), $\beta(\tau)$ is split into an asymptomatic, pre-symptomatic and symptomatic part. In eq(4), the assumption is made that the three all have the same shape, which is therefore equal to the shape of $\omega(\tau)$. Later (line 365), the values of x_a and x_p are assumed to be 1. As a result, eq (5) will be simplified to $\beta(\tau) = \beta_S(\tau)$.

Response: We thank the reviewer for this comment. Based on eq (3-5), the shape of $\beta(\tau)$ may not be equal to the shape of $\beta_s(\tau)$ because the cumulative distribution function of the incubation period $s(\tau)$ was also involved in eq (5). Therefore, the shape of $\beta_s(\tau)$ may not be same as the shape of $\omega(\tau)$. If x_a and x_p are assumed to be 1, eq (5) will be simplified to $\beta(\tau) = \beta_s(\tau)$. As a result, the shape of $\beta_s(\tau)$ is same as the shape of $\omega(\tau)$.

3. Describing what I read and understand: In eq(6), $\beta_S(\tau)$ is defined as a sum of five contributions for the five locations. In eq(9) the contributions are split into a contact rate (independent of time) and a probability of transmission $p_S^{ci}(\tau)$ (dependent of time). In line 348, the assumption is made that the time-dependent probabilities between the settings are only different by a scaling factor. Therefore, $p_S^{ci}(\tau)$ all have the same shape as $\omega(\tau)$.

Response: We appreciate your insight. Indeed, when $x_a = x_p = 1$, $p_S^{ci}(\tau)$ all have the same shape as $\omega(\tau)$. Determining $p_S^{ci}(\tau)$ requires R_0 based on eq (1), along with c_i and y_{ci} based on eq (6) and eq (8) in the revised Supplementary Materials. It's important to note that when the x_a and x_p are changed, the $p_S^{ci}(\tau)$ might not have the same shape as $\omega(\tau)$.

4. Describing what I read and understand: from all the above, all that remains is $R = \sum_i [c_i * y_{ci} * \int p_S^{ci}(\tau) d\tau]$. In words: the reproduction number is a sum of the contribution of the five settings i , which ONLY differ by their mean contact rate c_i and their

mean probability of transmission per contact y_{ci} , relative to that probability in setting 1 (the integral).

Response: We thank the reviewer for this comment. The reproduction number is indeed a sum of the contribution of the five settings i , which solely differ by their mean contact number per day c_i and their scaling factor y_{ci} . This scaling factor represents the ratio of probability of transmission per contact in contact location i over the dwelling location.

5. Concluding from the above descriptions: I can understand how you can estimate c_i and y_{ci} from the numbers in Table S3. What I do not understand is how you can use these to calculate the contributions to R , in the absence of NPI. What data are available, and what values do change in the above model?

Response: We are sorry for the unclarity. As we stated earlier, we calculated the transmission contribution and SSE risk without NPIs based on the following information: (1) the contact number per day of tracing in Figure 2 can be treated as the normal contact pattern (i.e., without NPIs); (2) The transmission probabilities under different locations (Figure S6) were our interest. We obtained them under $R_0=10$ (i.e., without NPIs) and the normal contact pattern. Thus, they can be treated as the biological nature of SARS-CoV-2. The transmission contribution was obtained based on eq (13) (according to the latest version of the appendix numbering) with the normal contact pattern and the scaling factor y_{ci} . Note that y_{ci} was obtained based on the contacts being positive among all the contacts in the corresponding locations. The dynamic zero COVID-19 policy may have limited effects on them. The key is that we considered the contacts before isolations or quarantine as the normal contact pattern, which can serve as a baseline of human behavior in Beijing for other infectious disease studies. Considering the implemented dynamic zero COVID policy, we adjusted the contact patterns based on intra-city mobility data from Beijing and the conclusions did not vary significantly (Line 154-158 of Page 4 in the revised Main Text and Figure S7-S8 in revised Supplementary Materials). If we update the c_i and $p_{S^ci}(\tau)$ according to the specific NPIs, then the eq (13) would give the results under NPIs scenario.

Line 361 “numerically solving $\beta_S(\tau)$ from the inferred generation interval distribution”: according to the above observation #1, the shapes of $\beta_S(\tau)$ and $\omega(\tau)$ are identical, so what is there to solve? The values in lines 363-365 are not needed.

Response: We appreciate the reviewer for this comment. When $x_a = x_p = 1$, the shapes of $\beta_S(\tau)$ and $\omega(\tau)$ are identical. Therefore, the values of x_a and x_p are necessary. R_0 was also required based on eq (1). It's important to note that if different values are set for x_a and x_p , all other parameters mentioned in Line 363-365 (numbering in the previous version) are needed. This modelling framework can be adapted for other infectious diseases with modifications to the epidemiological parameters. Hence, we provided this modeling explanation

for its generality. We have revised this part in new version of Supplementary Materials (Line 257-266 of Page 7).

Line 365 “ R_0 was set to 10”: why would you need a value for relative contributions?

Response: We appreciate the reviewer for this comment. R_0 was used to determine the $\beta_S(\tau)$ and calculate the basic reproduction number for each location. The basic reproduction number for each location was the mean of secondary cases distribution, which is directly related to the SSE risk. Furthermore, comparing the numerical values of transmissions across different locations can provide a more intuitive and meaningful for understanding of the varying degrees of transmission across different locations.

Lines 365-368 “Subsequently, we could determine the proportion of infected contacts...”: which parameter in the model do you calculate here? And how do you calculate y_{ci} ?

Response: We apologize for any confusion. In the model, the parameter y_{ci} is indeed essential. To obtain them, we first determined the proportion of infected contacts under each contact location, calculated as the observed infected contacts divided by all the contacts in the corresponding location (as presented in Table S3). Specifically, the proportions are 12.7% for dwelling, 1.2% for workplace, 0.4% for cospace-time interaction, 0.5% for community settings and 4.3% for unknown settings. Using dwelling as reference, $y_{c1}=1$, $y_{c2}=1.2\%/12.7\%=0.094$, $y_{c3}=0.4\%/12.7\%=0.031$, $y_{c4}=0.5\%/12.7\%=0.039$, $y_{c5}=4.3\%/12.7\%=0.339$. We have revised this part in the new version of Supplementary Materials (Line 259-266 of Page 7).

Lines 368-369 “Furthermore, we derived the distribution for c_i ” and further: the model is all about means and expectations, which is obvious given that the calculations are about the reproduction number. Therefore, I do not understand why there is any mention of distributions here. I also do not understand the difference between c_i and C_i (apart from a factor 20??). What is really puzzling is the phrase in line 386 “contact number at time τ ”, which suggests that c_i is not a constant.

Response: We appreciate the reviewer's feedback. The distribution is indeed crucial for determining SSE risk. As we mentioned earlier in the response letter, we used the distribution of contact number per day to derive the distribution of secondary cases and subsequently calculated SSE risk for each location. The SSE was defined as the secondary cases of a case exceeding the 99th percentile $Z(R_0)$ of a Poisson distribution with R_0 as the mean (Lloyd-Smith *et al*, Nature, 2005). The probability of SSE was determined as the integral proportion of the secondary cases exceeding $Z(R_{loc})$ based on the distribution of secondary cases in each location. Sorry for the confusing notations. C_i and c_i represents the contact number per day. To approximate the integral, we used $1/M=0.05$ day as time unit. Sensitivity analysis showed that other values of M

won't change our results. For "contact number at time tau", we originally intended to convey that c_i can vary over time, but it is assumed to be constant over time due to the lack of more detailed information. We have revised this part accordingly.

Lines 388-397: At this point I'm really lost, and I don't follow how the observed contributions are translated to contributions under the 'biological nature of SARS-CoV-2', i.e. without NPIs

Response: We are sorry for the unclarity. Our interest lied in obtaining the transmission probabilities under different locations (as shown in Figure S6). An intensive contact tracing effort was conducted in Beijing, identifying 2,230 cases and 220,878 contacts. Notably, our dataset included contacts who tested negative for SARS-CoV-2, providing a unique aspect. Moreover, the contacts before the quarantine or isolation of cases were analyzed. These allow us to capture the social contact patterns, approximating normal contact patterns (Figure 2). The dynamic infectiousness under different locations (i.e., Figure S6) were determined under normal contact pattern and $R_0=10$ (i.e., without NPIs), which can be treated as biological nature of SARS-CoV-2.

We obtain the transmission contribution without NPI using the normal contact pattern and $p_{S^c1}(\tau)$ based on the right-hand side of eq (13) [i.e., $(E(c_1) + E(c_2)y_{c2} + E(c_3)y_{c3} + E(c_4)y_{c4} + E(c_5)y_{c5}) \int_0^\infty p_S^{c1}(\tau) (p_a x_a + (1 - p_a)(1 - s(\tau))x_p + (1 - p_a)s(\tau)) d\tau$]. Our logic is that this describe the physical mechanisms of the SARS-CoV-2 transmission in the real world. It can be applicable in more complex situations, for example, the changing contact pattern over time due to NPIs and/or the modified $p_{S^c1}(\tau)$ due to the medical treatments.

We revised these 3.2 section to improve the readability. We sincerely thank the reviewer once again for the valuable comments, which have greatly improved the quality of our work.

REVIEWERS' COMMENTS

Reviewer #3 (Remarks to the Author):

Thank you very much for the more detailed explanation. I am happy with the current version and have only two minor remarks, both related to the fact that numbers of contacts are natural numbers (integers), not continuous:

Line 69: numbers should be integers (and I think the interval is not a confidence interval)

Lines 100-108: gamma distributions are fitted, but I think that negative binomial distributions would be more appropriate.

Manuscript ID: NCOMMS-23-58901B

General Response:

We sincerely appreciate the time and effort that the reviewers dedicated to providing feedback on our manuscript. Their insightful comments and suggestions have been invaluable in improving the quality of our paper. We have carefully incorporated all the suggestions made by the reviewers, with particular attention to the minor remarks. These changes are highlighted in the manuscript.

Please find our responses below, in blue, for a point-by-point response to the reviewers' comments and concerns. All page and line numbers refer to the revised manuscript with tracked changes.

REVIEWER COMMENTS

Reviewer #3 (Remarks to the Author):

Thank you very much for the more detailed explanation. I am happy with the current version and have only two minor remarks, both related to the fact that numbers of contacts are natural numbers (integers), not continuous:

Response: We thank the reviewer for the positive feedback on our revised manuscript. We greatly appreciate the kind words and are sincerely grateful for your valuable comments, which have significantly improved the quality of our work. Thank you once again for your time and effort.

Line 69: numbers should be integers (and I think the interval is not a confidence interval)

Response: We thank the reviewer for this suggestion. We apologize for the oversights. The interval should indeed be the IQR. We have made this revision in the latest version of the manuscript (Line 82, Page 3). Thank you for bringing this to our attention.

Lines 100-108: gamma distributions are fitted, but I think that negative binomial distributions would be more appropriate.

Response: We thank the reviewer for this suggestion. We agree that negative binomial distributions would be more appropriate to characterize the count data. Considering the dependence of contact numbers over the tracing window for a primary case, the average of contact number per day for each case (which is continuous) was employed for the distribution fitting. Therefore, the gamma distribution was used. In general, negative binomial distributions can be viewed as a Poisson(λ) distribution, where λ is also a random variable, distributed as a gamma distribution with shape r and scale $\theta = (1 - p)/p$. The Gamma distribution was chosen also based on its statistical properties, particularly with regard to scaling. Thank you for your insightful comment.